# Society of Agents: Regret Bounds of Concurrent Thompson Sampling

**Yan Chen**[1*]    **Perry Dong**[2*]    **Qinxun Bai**[3]    **Maria Dimakopoulou**[4]    **Wei Xu**[3]    **Zhengyuan Zhou**[5,6†]

[5]Arena Technologies    [1]Duke    [3]Horizon Robotics    [6]NYU Stern    [4]Spotify    [2]UC Berkeley

yan.chen@duke.edu    perrydong@berkeley.edu

{qinxun.bai, wei.xu}@horizon.ai    mdimakopoulou@spotify.com    zzhou@stern.nyu.edu

## Abstract

We consider the concurrent reinforcement learning problem where $n$ agents simultaneously learn to make decisions in the same environment by sharing experience with each other. Existing works in this emerging area have empirically demonstrated that Thompson sampling (TS) based algorithms provide a particularly attractive alternative for inducing cooperation, because each agent can independently sample a belief environment (and compute a corresponding optimal policy) from the joint posterior computed by aggregating all agents' data , which induces diversity in exploration among agents while benefiting shared experience from all agents. However, theoretical guarantees in this area remain under-explored; in particular, no regret bound is known on TS based concurrent RL algorithms.

In this paper, we fill in this gap by considering two settings. In the first, we study the simple finite-horizon episodic RL setting, where TS is naturally adapted into the concurrent setup by having each agent sample from the current joint posterior at the beginning of each episode. We establish a $\tilde{O}(HS\sqrt{\frac{AT}{n}})$ per-agent regret bound, where $H$ is the horizon of the episode, $S$ is the number of states, $A$ is the number of actions, $T$ is the number of episodes and $n$ is the number of agents. In the second setting, we consider the infinite-horizon RL problem, where a policy is measured by its long-run average reward. Here, despite not having natural episodic breakpoints, we show that by a doubling-horizon schedule, we can adapt TS to the infinite-horizon concurrent learning setting to achieve a regret bound of $\tilde{O}(DS\sqrt{ATn})$, where $D$ is the standard notion of diameter of the underlying MDP and $T$ is the number of timesteps. Note that in both settings, the per-agent regret decreases at an optimal rate of $\Theta(\frac{1}{\sqrt{n}})$, which manifests the power of cooperation in concurrent RL.

## 1   Introduction

In concurrent reinforcement learning [26], many agents interact with (different instances of) the same environment simultaneously and share data with each other in order to jointly learn an optimal policy for the environment. This learning scenario has important applications for large-scale automated sequential decision making systems. For instance, in recommendation systems [1]) (e.g. movie recommendation, personalized ads recommendation, personalized news/content recommendation), a company often has a large number of servers – sometimes located in different geographical regions – that serve the entire population of the customer base. As such, an incoming customer is randomly routed to a server (i.e. an agent), which learns to make recommendation decisions in conjunction with

---

*Equal contributions.

†Correspondence to Zhengyuan Zhou (`z@arena-ai.com`).

36th Conference on Neural Information Processing Systems (NeurIPS 2022).

all the other servers, and the performance judgement is made on the entire set of servers. Another example is in concurrent personalized promotions, where each agent is actively learning to provision personalized promotions on a product, along with other agents, each of which is performing the same personalized promotions task on a similar product. As surprising as it sounds, this example has already been done in practice, where Arena Technologies[†], a cutting-edge enterprise AI solution provider, deploys sophisticated reinforcement learning agents to provision personalized promotions/pricing to an ensemble of beverages and other consumer packaged goods. A third example – which exists as a concept but not yet happened– of concurrent RL is controlling a team of autonomous vehicles, which, as succinctly summarized in [9], concerns a scenario where "each agent manages a single vehicle, and again, the agents learn from each other as data is gathered. The goal could be to optimize a combination of metrics, such as fuel consumption, safety, and satisfaction of transportation objectives. Exploratory actions play an important role, and structured diversity of experience may greatly accelerate learning." This application is particularly relevant and presents an unique opportunity for the emerging era of autonomous driving. Other concurrent RL applications include robotics ([10], [26]) and biology([27]).

An important opportunity in the concurrent RL setup –in comparison to the traditional single-agent RL setup – lies in coordinated exploration: since those agents can share their own learning experience with others, the team as a whole should achieve better overall regret performance (or equivalently better per-agent regret performance) if they intelligently coordinate the exploration efforts. In particular, different agents may want to diversify their exploratory actions so as to quickly reduce the team's uncertainty over the landscape of the environment in which they all operate. By sharing the resulting data with each other, there is hope that the team would–in a environment-interaction time frame much less than that from a single-agent setting–achieve the same level of performance.

When it comes to exploration, a substantial amount of literature has been devoted to designing (provably) efficient algorithms in the single-agent RL setting. One prominent class of algorithms, based on upper confidence bounds, is known as UCRL (and its variants); (see[13] [4]) and many references citing them and cited by them. The UCRL family of algorithms embodies the important philosophy of "optimism in face of uncertainty" in their design and enjoys optimal theoretical regret guarantees (along with elegant analysis) in various settings, including both finite-horizon episodic RL and infinite-horizon average-reward RL. As such, it is perhaps not surprising that UCRL algorithms are the first family of algorithms that have been adapted to the concurrent RL settings [11] and [23]. In particular, concurrent UCRL has been analyzed for the sample complexity performance measure (i.e. how many samples are needed to learn an $\epsilon$-optimal policy) under both finite action space setting and infinite action space setting. More specifically, [11] provided a high-probability bound of $\tilde{O}(\frac{S^2A}{\epsilon^3} + \frac{SAn}{\epsilon})$ for the sample complexity with $n$ agents interacting in the environment. Their algorithm was extended from MBIE(see [28]), with a single agent performing concurrent RL across a set of $n$ infinite-horizon MDPs. The results there show that with sharing samples from copies of the same MDP a linear speedup in the sample complexity of learning can be achieved. But no regret bound was derived there for concurrent RL. [23] designed an algorithm which extended concurrent exploration to continuous state-action spaces and allowed smooth generalization between concurrent MDPs under infinite horizon setting. Likewise, the authors also explored the scenario where one single agent interacts with concurrent MDPs at the same time. Additionally, they define an error metric, called TCE (total cost of exploration), and showed an $\tilde{O}((\frac{1}{\epsilon} + n)\mathcal{N}(d))$ bound for TCE, where $\mathcal{N}(d)$ is a constant representing the covering number of the MDP. This is sufficient for an $\tilde{O}((\frac{1}{\epsilon^2} + \frac{n}{\epsilon})\mathcal{N}(d))$ bound for sample complexity.

However, these sample-complexity guarantees notwithstanding, concurrent UCRL algorithms suffer from the critical disadvantage of no coordinated exploration: since the upper confidence bounds computed by aggregating all agents' data are the same the entire team, each agent would follow the exact same policy, thereby yielding no diversity in exploration. This point has been observed in [9, 8], which has advocated for designing concurrent RL algorithms based on Thompson sampling [20],[21], [29], [14] [17],[18], also known as posterior sampling reinforcement learning (PSRL). Unlike UCRL, PSRL manifests the philosophy of "probabilistic optimism in face of uncertainty", where the agent is uncertain about the transition probabilities and reward functions, and hence at the start of each episode, the agent would sample an MDP from the current posterior distribution based on data

---

[†]See https://www.arena-ai.com/ for more information.

gathered up to that time and makes decisions according to the policy that is optimal with respect to the sampled MDP. [6] and [22] have provided good motivating examples for empirical evidence.

In fact, not only PSRL is well-known to outperform UCRL empirically in single-agent RL setting, it also – by its very design – lends itself well to concurrent RL settings: each agent can sample from a posterior distribution and explores accordingly, which induces diversity in a coordinated way. Building on this insight, [9] and [8] proposed sophisticated Thompson sampling variants that demonstrated good empirical performance. Despite these remarkable empirical results, to date, no theoretical guarantee has been provided on any concurrent Thompson sampling algorithm, simple or sophisticated. Our goal in this paper is to make initial inroads on the theoretical front and attempt to prove the first bounds on simple-but-natural concurrent PSRL algorithms.

### 1.1 Our Contributions

We establish per-agent regret guarantees for concurrent PSRL algorithms in two settings: one in finite-horizon episodic setting and the other in infinite-horizon average reward setting. In the first setting, TS is naturally adapted into the concurrent setup by having each agent sample from the current joint posterior at the beginning of each episode. We establish a $\tilde{O}(HS\sqrt{\frac{AT}{n}})$ per-agent regret bound, where $H$ is the horizon of the episode, $S$ is the number of states, $A$ is the number of actions, $T$ is the number of episodes and $n$ is the number of agents. This regret bound further improves to $\tilde{O}(H\sqrt{\frac{SAT}{n}})$ under the special Dirichlet prior. Note that when $n = 1$, both regret bounds are tight(see [16], [21]). These two regret bounds further highlight that the per-agent regret decreases at an optimal rate of $\Theta(\frac{1}{\sqrt{n}})$ and hence manifests the power of cooperation in concurrent RL: as a team, the total regret achieved is $\tilde{O}(HS\sqrt{nAT})$ (and $\tilde{O}(H\sqrt{nSAT})$ under Dirichlet prior), which is minimax optimal, because that is the optimal regret performance (up to log factors) achievable by a single-agent for $nT$ episodes that can adapt at each episode[†].

In the second setting, we consider the infinite-horizon RL problem, where a policy is measured by its long-run average reward. Here, despite not having natural episodic breakpoints, we show that by a doubling-horizon schedule, we can adapt TS to the infinite-horizon concurrent learning setting to achieve a regret bound of $\tilde{O}(DS\sqrt{ATn})$, where $D$ is the standard notion of diameter of the underlying MDP and $T$ is the number of timesteps. Here, we again see that the per-agent regret decreases at an optimal rate of $\Theta(\frac{1}{\sqrt{n}})$ and the fact that a single learning trajectory does not prevent the team of agents from efficient concurrent reinforcement learning. Due to space limitation, we'll include simulation plots in the appendix.

## 2 Problem Formulation

We consider the Bayesian regret bound of concurrent Thompson Sampling of Markov decision process in finite-horizon episodic setting and infinite-horizon setting. In both settings, we provide bounds on the general prior distributions and Dirichlet prior distributions for concurrent Thompson Sampling of the MDPs.

### 2.1 Finite-Horizon Episodic Setting

Formally, we consider the problem of learning to optimize a random (parallel) finite-horizon episodic Markov decision process $M^* = (\mathcal{S}, \mathcal{A}, R^*, P^*, H, \rho)$ over repeated episodes of interaction. Here $\mathcal{S} = \{1, 2, \ldots, S\}$ is the state space, $\mathcal{A} = \{1, 2, \ldots, A\}$ is the action space, $H$ is the horizon of each episode, and $\rho$ is the initial state distribution for each episode. At the start of each episode $k(k \geq 1)$, each agent will receive the history across all the agents before period $k$. And at the start of episode 1, all the agents don't have any information regarding the history. For every period $h = 1, 2, \ldots, H$ within an episode, each agent $p$ observes state $s_h^p \in \mathcal{S}$, selects action $a_h^p \in \mathcal{A}$, receives a reward $r_h \sim R^*(s_h^p, a_h^p)$, where the reward functions $r_h$ are assumed to be in $[0, 1]$, and transitions to a new state $s_{h+1}^p \sim P^*(s_h^p, a_h^p)$. At time $t$, agent $p$ chooses a policy $\pi^p$, which is

---

[†]That is, one can equivalently think of the concurrent setting as a single agent making decisions but only being able to adapt every $n$ time episodes. As such, the team's performance as a whole in $T$ episodes will never be better than for the single agent in $nT$ episodes.

a mapping from the information that she has, i.e. $\mathcal{H}_t^p$, to action $a \in \mathcal{A}$ chosen by agent $p$. For each agent $p$, MDP $M$ and policy $\pi^p$, the state-action value function for each period $h$ is defined as $Q_{\pi^p,h}^M(s,a) = \mathbb{E}_{\pi^p,M}[\sum_{j=h}^H \bar{r}^M(s_j,a_j)|s_h = s, a_h = a]$, where $\bar{r}^M(s,a) = \mathbb{E}[r|r \sim R^M(s,a)]$. The subscript $\pi^p$ indicates that the actions of agent $p$ over periods $h+1, \ldots, H$ are chosen according to policy $\pi^p$. Let $V_{\pi^p,h}^M(s) := Q_{\pi^p,h}^M(s, \pi^p(s,h))$. We say a policy $\pi^M$ is optimal for MDP $M$ if $\pi^M \in \arg\max_\pi V_{\pi,h}^M(s)$ for all $s \in \mathcal{S}$ and $h = 1, 2, \ldots, H$.

Let $\mathcal{H}_t = \cup_{p=1}^n \mathcal{H}_t^p$ denote the history of observations made by time $t$ (prior to and including time step $t$) across all the agents $p = 1, 2, \ldots, n$, where $\mathcal{H}_t^p$ is the history of agent $p$ by time step $t$ (include time $t$), with $t = 1, 2, \ldots, T$ and $T$ is the total time horizon. So $t = (k-1)H + h$ is the time step at period $h$ within episode $k$. We use $s_{kh}^p$ to denote the state of agent $p$ at time $t = (k-1)H + h$, i.e. episode k period $h$. The RL algorithm we consider here is a deterministic sequence $\{\pi_k : k = 1, 2, \ldots\}$ of functions, each mapping $\mathcal{H}_{(k-1)H}$ (i.e. the history of observations prior to episode $k$) to a probability distribution $\pi_k(\mathcal{H}_{(k-1)H})$ over policies, from which each agent samples a policy $\pi_{kp}$ for the $k$-th episode. The regret for agent $p$ in episode $k$ is $\Delta_{kp} := \sum_{s \in \mathcal{S}} \rho(s)(V_{\pi^*,1}^{M^*}(s) - V_{\pi_{kp},1}^{M^*}(s))$. The overall regret in episode $k$ is $\Delta_k := \sum_{p=1}^n \sum_{s \in \mathcal{S}} \rho(s)(V_{\pi^*,1}^{M^*}(s) - V_{\pi_{kp},1}^{M^*(s)})$, where $\pi^* = \pi^{M^*}$. The regret incurred up to time $T$ for policy $\pi$ is $\text{Regret}(T, \pi, M^*) = \sum_{k=1}^{\lceil T/H \rceil} \Delta_k$. The Bayesian expected regret for $M^*$ distributed according to the prior $\phi$ is defined as $\text{BayesRegret}(T, \pi, \phi) := \mathbb{E}[\text{Regret}(T, \pi, M^*)|M^* \sim \phi]$

## 2.2  Infinite-Horizon Setting

We follow the setting in [3] and consider the case where different agents can cooperate. As in [3] we consider communicating MDPs with finite diameter. We first define 'diameter', 'communicating' and other related terms.

**Definition 2.1.** A deterministic policy $\pi : \mathcal{S} \to \mathcal{A}$ is a mapping from state space $\mathcal{S}$ to action space $\mathcal{A}$.

**Definition 2.2.** Diameter $D(\mathcal{M})$ of an MDP $\mathcal{M}$ is defined as the minimum time required to go from one state to another in the MDP using some deterministic policy: $D(\mathcal{M}) = \max_{s \neq s', s, s' \in \mathcal{S}, \pi:\mathcal{S} \to \mathcal{A}} T_{s \to s'}^\pi$. Here $T_{s \to s'}^\pi$ is the expected number of steps it takes to reach state $s'$ when starting from state $s$ and using policy $\pi$.

**Definition 2.3.** An MDP is communicating if and only if it has a finite diameter.

**Definition 2.4.** The gain $\lambda^\pi(s)$ of a policy $\pi$, from starting state $s_1 = s$, is defined as the infinite horizon undiscounted average reward given by $\lambda^\pi(s) = \mathbb{E}[\lim_{T \to \infty} \frac{1}{T} \sum_{i=1}^T r(s_t, \pi(s_t))|s_1 = s]$, where $s_t$ is the state reached at time $t$, by executing policy $\pi$.

**Lemma 2.1.** (Optimal gain for communicating MDPs). For a communicating MDP $\mathcal{M}$ with diameter $D$ we have: 1. ([24] Theorem 8.1.2, Theorem 8.3.2) The optimal gain $\lambda^*$ is state independent and is achieved by a deterministic stationary policy $\pi^*$, i.e. there exists a deterministic policy $\pi^*$ such that $\lambda^* = \max_{s' \in \mathcal{S}} \max_\pi \lambda^\pi(s') = \lambda^{\pi^*}(s), \ \forall s \in \mathcal{S}$; 2. ([5], Theorem 4). The optimal gain $\lambda^*$ satisfies the following equations $\lambda^* = \min_{h \in \mathbb{R}^S} \max_{s,a} r(s,a) + P_{s,a}^T h - h_s = \max_a r(s,a) + P_{s,a}^T h^* - h_s^* \ \forall s$ where the vector $h^* = (h_s)_{s \in \mathcal{S}}$ satisfies the following optimality equation $h^* + \lambda^* \mathbf{e} = \max_{a \in \mathcal{A}}(r(s,a) + P_{s,a}^T h^*)$. Here $h^*$ is referred to as the bias vector of MDP $\mathcal{M}$ satisfying: $\max_s h_s^* - \min_s h_s^* \leq D$.

Given the above definitions and results, we now formally define the Reinforcement Learning problem we consider here. The process proceeds in time steps $t = 1, 2, \ldots, T$. For agent $p = 1, 2, \ldots, n$, each starts from a state $s_1^p$ at time step $t = 1$. In the beginning of each time step $t$, agent $p$ takes an action $a_t^p \in \mathcal{A}$ and observes the reward $r(s_t^p, a_t^p)$, where $s_t^p$ is the state of agent $p$ at time $t$, and also the next state $s_{t+1}^p \sim P(s_t^p, a_t^p)$. And again the reward function $r$ is assumed to be in $[0, 1]$. Here $r$ and $P$ are the reward function and the transition model respectively for a communicating $MDP$ $\mathcal{M}$ with diameter $D$. Each learning agent knows the state-space $\mathcal{S}$, the action space $\mathcal{A}$, as well as the rewards $r(s,a), \forall s \in \mathcal{S}, a \in \mathcal{A}$, for the underlying MDP, but not the transition model $P$ or the diameter $D$. The agent can use the past observations across all the agents to learn the underlying MDP model and decide future actions. The goal for each agent is to maximize total reward $\sum_{t=1}^T r(s_t^p, a_t^p)$, or equivalently minimize the total regret over a time horizon $T$, defined as $\mathcal{R}(T, \mathcal{M}, n) = Tn\lambda^* - \sum_{p=1}^n \sum_{t=1}^T r(s_t^p, a_t^p)$, where $\lambda^*$ is the optimal gain of MDP $\mathcal{M}$.

**Remark 2.1.** Unlike the finite-horizon case, we assume that we know the reward functions as in [3]. Since without loss of generality, as long as the reward functions are bounded, we can have the same bound on the regret. This is because the unknown transition probability is the main source of difficulty in the analysis of the regret bound.

**Notations** We use $\mathcal{S}$ to denote the state space, where $|\mathcal{S}| = S$, and $\mathcal{A}$ to denote the action space, with $|\mathcal{A}| = A$. There are $n$ agents interacting with the environment. We use $r(s, a)$ to denote the reward related to state-action pair $(s, a)$. In finite horizon case, $H$ is the horizon of each episode, $\rho$ is the initial state distribution for each episode. $s_t^p$ is the state of of agent $p$ at time step $t$, and $a_t^p$ is the action that agent $p$ takes at time step $t$, where $p = 1, 2, \ldots, n$. In finite horizon case, we use $P(s, a)$ to refer to the transition probability vector related to state-action pair $(s, a)$, and use $P_{s,a}$ under the infinite horizon case. The policy $\pi^p$ is a mapping from the information set that agent $p$ has access to, to an action $a$ in the action space. Under finite horizon, we use $s_{kh}^p$ to indicate the state of agent $p$ in episode $k$ period $h$, and $a_{kh}^p$ to indicate the action that agent $p$ takes in episode $k$ period $h$. We use $P^*$ to denote the transition probability of the true MDP, $\lambda^*$ to denote the optimal gain of the true MDP under infinite horizon case. $\tilde{\lambda}_k$ is the optimal gain of the sampled MDP in epoch $k$ under infinite horizon case. $D$ is the upper bound of the diameter of the MDP considered. For finite horizon case, $\mathcal{H}_t^p$ denotes the information gathered by agent $p$ before time step $t$, and $\mathcal{H}_t = \cup_{p=1}^n \mathcal{H}_t^p$ is the history of observations prior to time step $t$ across all agents. For infinite horizon case, $\mathcal{F}_k^p$ is the information of agent $p$ within and before epoch $k$, and $\mathcal{F}_k = \cup_{p=1}^n \mathcal{F}_k^p$ is the information within and before epoch $k$ across all agents.

## 3 Finite-Horizon Episodic Case

**Algorithm** Here we consider a concurrent version of the PSRL in [21]. At the beginning of each episode, each agent samples an MDP independently, where each one computes the optimal policy for their sampled MDP, and proceeds with this policy during the rest of the episode. Then they gather and update information at the end of each episode (See Algorithm 1). There are $n$ agents interacting in the environment, each episode is of length $H$, and the number of total time steps is $T$. So there are $K = \lceil T/H \rceil$ episodes in the environment.

**Intuition for $\tilde{O}(\frac{1}{\sqrt{n}})$ per-agent regret bound** We obtain an $\tilde{O}(\sqrt{Tn})$ Bayesian regret bound for finite-horizon case, with $\tilde{O}(HS\sqrt{ATn})$ for general prior distribution, and $\tilde{O}(H\sqrt{SATn})$ for Dirichlet prior distribution, where $n$ is the number of agents interacting in the environment. Note that in the single-agent case, previous literature shows such regret bounds, ($\tilde{O}(S\sqrt{AT})$ for general prior in [16] and $\tilde{O}(\sqrt{SAT})$ for Dirichlet prior in [21]). This indicates that when $n$ agents interactively learn in the same unknown environment over a time horizon of T, the bound scales in a non-linear way as if a single-agent learn over a time horizon of $T' = nT$, which suggests the optimality of our bounds here with respect to the number of agents $n$. This is also a reflection of the power of concurrent learning within concurrent RL.

**Improvement of Bayesian Regret Under Dirichlet Prior Distribution** The key analysis from $\tilde{O}(HS\sqrt{ATn})$ to $\tilde{O}(H\sqrt{SATn})$ follows the technical notes in [21], where we borrow a technical lemma on Gaussian-Dirichlet concentration bound ([19]). With this, we are able to show that the regret contribution from the transition probability estimate can concentrate at a rate independent of $S$. Finally we use a Sub-Gaussian tail bound to make the improvement. The proof idea follows [21].

**Decomposition of the Regret:** Note that we have

$$\text{BayesRegret}(T, \pi, \phi) = \mathbb{E}[\text{Regret}(T, \pi, M^*)|M^* \sim \phi]$$

$$= \sum_{k=1}^{\lceil T/H \rceil} \sum_{p=1}^{n} \mathbb{E}[\sum_{s \in \mathcal{S}} \rho(s)(V_{\pi^*,1}^{M^*}(s) - V_{\pi_{kp},1}^{M^*}(s))|M^* \sim \phi]$$

And we can write

$$V_{\pi^*,1}^{M^*}(s) - V_{\pi_{kp},1}^{M^*(s)} = \underbrace{V_{\pi^*,1}^{M^*}(s) - V_{\pi_{kp},1}^{M_{kp}}(s)}_{\Delta_{kp}^{\text{opt}}} + \underbrace{V_{\pi_{kp},1}^{M_{kp}}(s) - V_{\pi_{kp},1}^{M^*}(s)}_{\Delta_{kp}^{\text{conc}}}$$

---

**Algorithm 1:** Concurrent PSRL

---

**Data:** prior distribution $\phi, \mathcal{H}_0^p = \emptyset, \forall p \in [n]$

https://www.overleaf.com/project/628438b2462076a6e40eafa2 **for** *episode* $k = 1, 2, \ldots$ **do**

    **for** $p = 1 : n$ **do**

        Sample MDP $M_{kp} \sim \phi(\cdot | \mathcal{H}_{(k-1)H})$;

        Compute $\mu_{kp} \in \arg\max_\mu V_{\mu,1}^{M_{kp}}$;

        **for** *period h=1,…,H* **do**

            take action $a_{kh}^p = \mu_{kp}(s_{kh}^p, h)$;

            update $\mathcal{H}_{(k-1)H+h}^p = \mathcal{H}_{(k-1)H+h-1}^p \cup \{s_{kh}^p, a_{kh}^p, r_{kh}^p, s_{k(h+1)}^p\}$

        **end**

    **end**

    $\mathcal{H}_{kH} = \cup_{p=1}^n \mathcal{H}_{kH}^p$;   Update posterior distribution $\phi(\cdot | \mathcal{H}_{kH})$;

**end**

---

Note that conditioned upon any data $\mathcal{H}_{(k-1)H}$, the true MDP $M^*$ and the sampled $M_{kp}$ are identically distributed. This means that $\mathbb{E}[\Delta_{kp}^{\mathrm{opt}}] \leq 0$ for all $k, p$. Therefore, we just need to bound $\sum_{k=1}^{\lceil T/H \rceil} \sum_{p=1}^P \mathbb{E}[\Delta_{kp}^{\mathrm{conc}} | \mathcal{H}_{(k-1)H}]$. For the convenience of notation, we write $V_{kh}^{kp} = V_{\pi_{kp},h}^{M_{k,p}}$. Now we rewrite the regret via the Bellman operator, with $w^R(x) = \bar{r}_{kp}(x) - \hat{r}_k(x), w_h^P(x) = (P_{kp}(x) - \hat{P}_k(x))^T V_{k,h+1}^{kp}$:

$$
\begin{aligned}
\mathbb{E}[\Delta_{kp}^{\mathrm{conc}} | \mathcal{H}_{(k-1)H}] &= \mathbb{E}[(\bar{r}_{kp} - \bar{r}^*)(x_{k1}^p) + P_{kp}(x_{k1}^p)^T V_{k2}^{kp} - P^*(x_{k1}^p) V_{k2}^* | \mathcal{H}_{(k-1)H}] \\
&= \mathbb{E}[(\bar{r}_{kp} - \bar{r}^*)(x_{k1}^p) + (P_{kp}(x_{k1}^p) - \hat{P}_k(x_{k1}^p))^T V_{k2}^{kp} + \mathbb{E}[(V_{k2}^{kp} - V_{k2})(s') | s' \sim P^*(x_{k1}^p)] | \mathcal{H}_{(k-1)H}] \\
&= \ldots \\
&= \mathbb{E}[\sum_{h=1}^H (\bar{r}_{kp}(x_{k1}^p) - \hat{r}^*(x_{k1}^p)) + \sum_{h=1}^H \{(P_{kp}(x_{kh}^p) - \hat{P}_k(x_{kh}^p)) V_{kh}^{kp}\} | \mathcal{H}_{(k-1)H}] \\
&\leq \mathbb{E}[\sum_{h=1}^H |w^R(x_{kh}^p)| + \sum_{h=1}^H |w_h^P(x_{k,h+1}^p)| \,| \mathcal{H}_{(k-1)H}]
\end{aligned}
$$

### 3.1 Bayesian Regret Bound under General Prior Distribution

**Theorem 3.1. (Bayesian Regret Bound Under General Prior Distribution)** Let $M^*$ be the true MDP distributed according to any prior distribution $\phi$. If the number of agents $n$ satsifies $n \leq O(S^2 AT \log(SAnT))$, Then the Bayesian regret for Algorithm (1) is bounded by

$$
\mathbb{E}[\mathrm{Regret}(T, \pi, \phi, n)] = \tilde{O}(HS\sqrt{ATn})
$$

**Proof Sketch:** We first define a confidence set $\mathcal{M}_k$ for each episode $k$ as in [4], so that this set contains both sampled MDP $M_k$ and true MDP $M^*$ with high probability. Then we decompose $\Delta_{kp}$ into $\Delta_{kp}\mathbf{1}(M_k \in \mathcal{M}_k, M^* \in \mathcal{M}_k) + \Delta_{kp}[\mathbf{1}(M_k \notin \mathcal{M}_k \text{ or } M^* \notin \mathcal{M}_k)]$. Then we use the fact that when condition on history $\mathcal{H}_{(k-1)H}$, we have $\mathbb{E}[\mathbf{1}\{M_k \notin \mathcal{M}_k\} | \mathcal{H}_{(k-1)H}] = \mathbb{E}[\mathbf{1}\{M^* \notin \mathcal{M}_k\} | \mathcal{H}_{(k-1)H}]$([16]), so that $\Delta_{kp}[\mathbf{1}(M_k \notin \mathcal{M}_k \text{ or } M^* \notin \mathcal{M}_k)]$ is bounded by $2H\mathbb{P}(M^* \notin \mathcal{M}_k)$. As for the first part, namely $\Delta_{kp}\mathbf{1}(M_k, M^* \in \mathcal{M}_k)$, we found that within the confidence intervals, $\Delta_{kp}$ can be controlled with the desired upper bound. And for $\Delta_{kp}[\mathbf{1}(M_k \notin \mathcal{M}_k \text{ or } M^* \notin \mathcal{M}_k)]$, since it's bounded by $2H\mathbb{P}(M^* \notin \mathcal{M}_k)$, we just need to show that $\mathbb{P}(M^* \notin \mathcal{M}_k)$ is small, which is indeed the case. We decompose the regret into two parts, with one part negative and the other part positive. So we focus on bounding the positive part. Then we rewrite the positive upper bound via the Bellman operator, where one part is related to the concentration of reward estimate, and the other part is related to the concentration of transition probability estimate. i.e. $\mathrm{BayesRegret}(T, \pi, \phi) = \mathbb{E}[\mathrm{Regret}(T, \pi, M^*) | M^* \sim \phi] = \sum_{k=1}^{\lceil T/H \rceil} \sum_{p=1}^n \mathbb{E}[\sum_{s \in \mathcal{S}} \rho(s)(V_{\pi^*,1}^{M^*}(s) - V_{\pi_{kp},1}^{M^*}(s)) | M^* \sim \phi]$

And we can write $V^{M^*}_{\pi^*,1}(s) - V^{M^*(s)}_{\pi_{kp},1} = \underbrace{V^{M^*}_{\pi^*,1}(s) - V^{M_{kp}}_{\pi_{kp},1}(s)}_{\Delta^{\text{opt}}_{kp}} + \underbrace{V^{M_{kp}}_{\pi_{kp},1}(s) - V^{M^*}_{\pi_{kp},1}(s)}_{\Delta^{\text{conc}}_{kp}}$. Conditioned upon data before the start of episode $k$, $\mathcal{H}_{(k-1)H}$, the true MDP $M^*$ and the sampled $M_k$ are identically distributed. This means that $\mathbb{E}[\Delta^{\text{opt}}_{kp}] \leq 0$ for all $k,p$. Therefore, we just need to bound $\sum_{k=1}^{\lceil T/H \rceil} \sum_{p=1}^{P} \mathbb{E}[\Delta^{\text{conc}}_{kp}|\mathcal{H}_{(k-1)H}]$. For the convenience of notation, we write $V^{kp}_{kh} = V^{M_{kp}}_{\pi_{kp},h}$. Now we rewrite the regret via the Bellman operator, with $w^R(x) = \bar{r}_{kp}(x) - \hat{r}_k(x), w^P_k(x) = (P_{kp}(x) - \hat{P}_k(x))^T V^{kp}_{k,h+1}$. Then by some simple derivation we find that $\mathbb{E}[\Delta^{\text{conc}}_{kp}|\mathcal{H}_{(k-1)H}] \leq \mathbb{E}[\sum_{h=1}^{H} |w^R(x^p_{kh})| + \sum_{h=1}^{H} |w^P_h(x^p_{k,h+1})||\mathcal{H}_{(k-1)H}]$. Based upon this, we prove that with high probability, both the true MDP and the sampled MDPs are in the confidence sets constructed here. And also conditional upon the event that both true MDP and sampled MDPs are in the confidence set, the total regret can be controlled.

### 3.2 Bayesian Regret Bound under Independent Dirichlet Prior Distribution

**Theorem 3.2. (Bayesian Regret Bound Under Dirichlet Prior Distribution)** Let $M^*$ be the true MDP distributed according to prior $\phi$ with independent Dirichlet prior over transition probabilities. Then the Bayesian regret for Algorithm (1) is bounded by

$$\text{BayesRegret}(T, \pi, \phi, n) \leq \tilde{O}(H\sqrt{SATn})$$

**Proof Sketch:** From [21], under Dirichlet prior, by a union bound at each $x^p_{kh} = (s,a)$ on $P(s,a)$ and $r(s,a)$, we have that for each $p$, with probability at least $1 - \frac{1}{nT}$,

$$\mathbb{E}[\sum_{h=1}^{H} \{|w^R(x^p_{kh})| + |w^P_h(x^p_{k,h+1})|\}|\mathcal{H}_{(k-1)H}] \leq \sum_{h=1}^{H} 2(H+1)\sqrt{\frac{2\log(4SAnT)}{\max\{N_k(x^p_{kh}) - 2, 1\}}}$$

By pigeonhole principle, when we sum over all the $k$ and $p$, we can decompose the right hand side of the above inequality according to the number of times that each state-action pair were visited. For those state-action pairs which were visited less than twice, since the reward function is within $[0, 1]$, the sum of these parts are upper bounded by $2SA + 1$. Lemma (D.5) guarantees that the regret from transition probability estimation error can be bounded at a rate independent of $S$, and Lemma (D.4) further provides a bound for the dominating Gaussian variable on the Dirichlet distribution. In this way we can get an improved bound of $\tilde{O}(H\sqrt{SAnT})$ from the case under general prior distribution. We leave the proof in the appendix.

## 4 Infinite-Horizon Case

**Algorithm** Algorithm 2 combines the idea of Thompson sampling and double epoch for infinite horizon case. Similar to the finite-horizon episodic case, the agents sample MDP independently at the beginning of each epoch. The difference is that unlike the finite-horizon case where each episode is of the same length, the infinite horizon adapt a "double epoch" strategy ([4], [3]), where we stop the current epoch and move on to the next one immediately as long as the number of visits to some state-action pair at least doubles. As in finite-horizon case, we can also get $\tilde{O}(\frac{1}{\sqrt{n}})$ per-agent Bayesian regret bound as well. For the similar reason as in the finite-horizon case, here we also have an improvement from $\tilde{O}(S\sqrt{ATn})$ to $\tilde{O}(\sqrt{SATn})$ if we condition on Independent Dirichlet prior distribution instead of on some general prior .

**From Finite-Horizon To Infinite Horizon: Doubling Epoch** [15] points out the necessity to have "phases" in infinite-horizon problem. In infinite-horizon case, if the reward of one state is slightly larger than the reward in the current state, the MDP tends to switch to the other state very easily. But for finite-horizon setting, an optimistic algorithm would eventually stop considering switches because eventually, the loss for switching would be assessed to be higher than the potential gain from switching. For this reason, recalculating the optimistic policy in each round may lead to the algorithm to switch states frequently, which can lead to linear regret in the end. As in [3], we borrow the idea of doubling epoch [4], where the next phase starts once the number of visits to some agent's state-action pair at least doubles. Since the total number of visits to any state-action pairs can be no more than $nT$, the number of epochs is bounded by some constant times $SA\log(nT)$.

## 4.1 Bayesian Regret Bound Under General Prior Distribution

---

**Algorithm 2:** Concurrent Infinite-Horizon Posterior Sampling MDP

---

**Data:** State space $\mathcal{S}$, Action space $\mathcal{A}$, starting state vector across all agents $s_1 \in \mathcal{S}^P$, reward function $r$, time horizon $T$, $\mathcal{F}_0^p = \emptyset$ for $p \in [n]$.

**for** *epochs* $k = 1, 2, \ldots$ **do**

    Sample transition probability vectors: For each $s, a$ generate sample probability vectors $Q_{s,a}^k$ as follows:

    **Posterior sampling**: use samples from the posterior sampling:

$$Q_{s,a}^k \sim \phi(\cdot | \mathcal{F}_{k-1})$$

    Compute policy $\tilde{\pi}^k$: as the optimal gain policy for the sampled MDP $\tilde{M}^k$ constructed using sample set $\{Q_{s,a}^k, s \in \mathcal{S}, a \in \mathcal{A}\}$.

    Execute policy $\tilde{\pi}^k$:

    **for** *time steps* $t = \tau_k, \tau_k + 1, \ldots$, *until* **_break epoch_** **do**

        **for** *agent* $p = 1, 2, \ldots, n$ **do**

            (This block can be implemented in parallel)

            Play action $a_t^p = \tilde{\pi}_k(s_t^p)$.

            Observe the transition to the next state $s_{t+1}^p$.

            Update $\mathcal{F}_k^p = \mathcal{F}_k^p \cup \{s_t^p, a_t^p, r_t^p, s_{t+1}^p\}$

        **end**

        If $N_{s_t^p, a_t^p}^{t+1} \geq 2 N_{s_t^p, a_t^p}^{\tau_k}$ for some p, (i.e. for some $(s_t^p, a_t^p)$), then set $\tau_{k+1} = t + 1$ and **break epoch**.

        $\mathcal{F}_k = \cup_{p=1}^n \mathcal{F}_k^p$   Update posterior distribution $\phi(\cdot | \mathcal{F}_k)$;

    **end**

**end**

---

**Theorem 4.1. (Bayesian Regret Bound Under General Prior Distribution)** For any communicating MDP $\mathcal{M}$ with $S$ states, $A$ actions, diameter $D$. Then for $T \geq \Omega(SA \log^4(SATn))$. Then

$$\text{BayesRegret}(T, \mathcal{M}, n, \phi) \leq \tilde{O}(DS\sqrt{ATn})$$

**Proof Sketch:** We first write the total regret as the sum of regrets over all the epochs. We focus on bounding the regret within any fixed epoch. Then we decompose the difference between the optimal gain of the true MDP and the reward functions as the sum of two parts: one part is the difference between the optimal gain of the true MDP and the optimal gain of the sampled MDP; the other part is the difference between the optimal gain of the sampled MDP and the reward function at some state-action pair. Then finally when we do a sum over all the epochs, we can get the desired regret bound. As defined before, $\mathcal{R}(T, \mathcal{M}, n) = Tn\lambda^* - \sum_{p=1}^n \sum_{t=1}^T r(s_t^p, a_t^p)$. $\lambda^*$ is the optimal gain of MDP $\mathcal{M}$. Algorithm (2) proceeds in epochs $k = 1, 2, \ldots, K$, where $K \leq SA \log(T)$. Now we define $R_k = (\tau_{k+1} - \tau_k) n \lambda^* - \sum_{t=\tau_k}^{\tau_{k+1}-1} \sum_{p=1}^n r_{s_t^p, a_t^p}$, where $R_k$ is the regret within epoch $k$. So we can write $R_k$ as $R_k = \sum_{t=\tau_k}^{\tau_{k+1}-1} \sum_{p=1}^n [\lambda^* - r_{s_t^p, a_t^p}] = \sum_{t=\tau_k}^{\tau_{k+1}-1} \sum_{p=1}^n [(\lambda^* - \tilde{\lambda}_k) + (\tilde{\lambda}_k - r_{s_t^p, a_t^p})]$, where $\tilde{\lambda}_k$ is the optimal gain of the sampled MDP $\tilde{M}^k$. From Lemma (2.1) we know that for any state $s$, optimal policy $\tilde{\pi}_k$ for communicating MDP $\tilde{M}^k$, action $a = \tilde{\pi}_k(s)$, $\tilde{\lambda}_k = r_{s,a} + \tilde{P}_{s,a}^T \tilde{h} - \tilde{h}_s$, where $\tilde{P}_{s,a} = Q_{s,a}^k$. So we have

$$\sum_{t=\tau_k}^{\tau_{k+1}-1} \sum_{p=1}^{n} (\tilde{\lambda}_k - r_{s_t^p, a_t^p}) = \sum_{t=\tau_k}^{\tau_{k+1}-1} \sum_{p=1}^{n} (\tilde{P}_{s_t^p, a_t^p} - P_{s_t^p, a_t^p} + P_{s_t^p, a_t^p} - \mathbf{1}_{s_t^p})^T \tilde{h}$$

$$= \sum_{t=\tau_k}^{\tau_{k+1}-1} \sum_{p=1}^{n} (\tilde{P}_{s_t^p, a_t^p} - P_{s_t^p, a_t^p})^T \tilde{h} + \sum_{t=\tau_k}^{\tau_{k+1}-1} \sum_{p=1}^{n} (P_{s_t^p, a_t^p} - \mathbf{1}_{s_t^p})^T \tilde{h}$$

$$= \sum_{t=\tau_k}^{\tau_{k+1}-1} \sum_{p=1}^{n} (Q_{s_t^p, a_t^p}^k - P_{s_t^p, a_t^p})^T \tilde{h} + \sum_{t=\tau_k}^{\tau_{k+1}-1} \sum_{p=1}^{n} (P_{s_t^p, a_t^p} - \mathbf{1}_{s_t^p})^T \tilde{h}$$

### 4.2 Bayesian Regret Bound Under Dirichlet Prior

**Theorem 4.2. (Bayesian Regret Bound Under Dirichlet Prior)** Let $\mathcal{M}$ be the true MDP distributed according to prior $\phi$ with any independent Dirichlet prior over transition probabilities. Then the Bayesian regret for Parallel Infinite-Horizon posterior sampling MDP algorithn 2 is bounded by

$$\text{BayesRegret}(T, \mathcal{M}, n, \phi) = \tilde{O}(D\sqrt{SATn})$$

**Proof Sketch:** We use Lemma (G.2) to prove the concentration bound between sampled transition probabilities and the expectation of the true probabilities conditional on the current data. And note that one key property of Thompson sampling is that conditional upon the current data, the true MDP and the sampled MDP are identically distributed.

### 4.3 Worse-case Bound Under Dirichlet Prior

**Theorem 4.3. (Worst-case Infinite-Horizon Regret Bound Under Dirichlet Prior)** With high probability, Algoithm 5 (provided in Appendix B) is bounded by

$$\mathcal{R}(T, \mathcal{M}, n) \leq \tilde{O}\left(DS\sqrt{ATn}\right)$$

## 5 Simulation

In this section, we describe our empirical study of concurrent Thompson Sampling on a Markov Decision Process with a Dirichlet transition function and a normal reward function for both finite-horizon and infinite-horizon cases. In particular, we train the concurrent Thompson Sampling agents on randomly sampled MDPs and evaluate the regrets of these agents at the end of each episode/epoch on a fixed number of environments sampled from the ground truth distribution. The simulation settings are moved to appendix due to page limit. Some of the experiment plots are shown in Figure(1a), (1b), (2a) and (2b).

## 6 Conclusion

We have extended PSRL from a single agent to multiple agents, where the agents can concurrent with each other to learn in parallel and share data with each other. We consider two settings: both finite-horizon episodic MDP and infinite-horizon MDP. For state space size $S$, action space size $A$, total accumulated time horizon $T$, episode horizon $H$, we show a $\tilde{O}(HS\sqrt{\frac{AT}{n}})$ per-agent Bayesian regret bound under general prior distribution; while especially for independent Dirichlet prior distribution, we show that a tighter bound can be achieved as $\tilde{O}(H\sqrt{\frac{SAT}{n}})$. Under infinite horizon case, for the general prior distribution, we achieve $\tilde{O}(DS\sqrt{\frac{AT}{n}})$ per-agent Bayesian regret bound; and for the infinite-horizon case under dirichlet prior distribution, we obtain a $\tilde{O}(D\sqrt{\frac{ATS}{n}})$ Bayesian regret bound, with $D$ being the diameter of MDP considered here. We have provided a theoretical upper bound for concurrent learning under posterior sampling RL setting, and the multiplicative constants in the bound were just properly chosen but not tuned in an optimistic way. For future work, it would

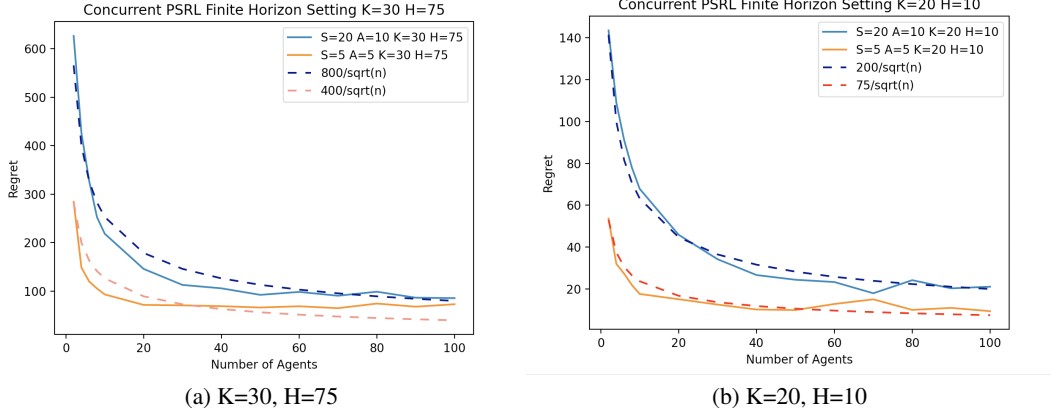

(a) K=30, H=75

(b) K=20, H=10

Figure 1: Bayesian regrets for Finite-Horizon Concurrent TS with Dirichlet Prior for Transition Probabilities and Gaussian prior for rewards. Solid curves are Bayesian regrets of our method vs number of concurrent agents. Dashed curves are reference curves given by our regret bounds $\Theta(\frac{1}{\sqrt{n}})$.

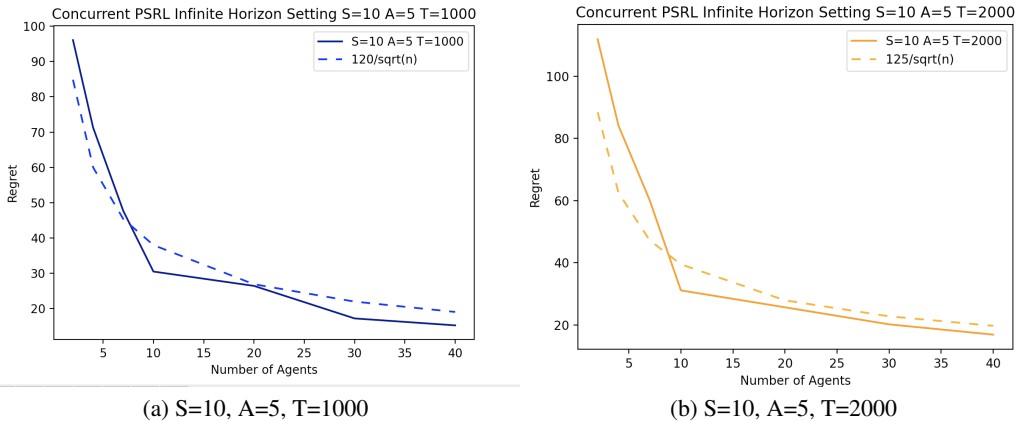

(a) S=10, A=5, T=1000

(b) S=10, A=5, T=2000

Figure 2: Bayesian regrets for Infinite-Horizon Concurrent TS with Dirichlet Prior for Transition Probabilities and Gaussian prior for rewards. Solid curves show Bayesian regrets of our method vs number of concurrent agents. Dashed curves are reference curves given by our regret bounds $\Theta(\frac{1}{\sqrt{n}})$.

be interesting to include communication cost ([2], [30]) and functional approximation in concurrent learning. One could evaluate this algorithm under model mis-specification ([33]). It is also valuable to explore model-free and worst-case bound settings([33], [32], [25], [7]). Another extension is to look at heterogeneous reward structure across agents, and compare the dynamic under the same reward structure setting for both PSRL and UCRL.

# 7 Acknowledgement

This work was supported in part by NSF award CCF-2106508. Zhengyuan Zhou acknowledges the generous support from Horizon Robotics faculty research fellowship and New York University's Center for Global Economy and Business faculty research grant. We would like to thank Pratap Ranade and Engin Ural for inspiring discussions on society of agents that have helped shape and push an ambitious vision of this research agenda, for which this work is only an initial step.

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
