# Supplementary Material

## A  Restatement of Algorithms

For the convenience of reference, we restate our algorithms under both finite-horizon case (Algorithm 1) and infinite-horizon case (Algorithm 2). In both finite and infinite horizon cases, we provided Bayesian regret bounds on both general prior and Dirichlet prior distributions. The reward function under finite-horizon case is set to be random and bounded within $[0, 1]$. The reward function under infinite-horizon case is set to bee known and also bounded within $[0, 1]$. However, the result can also be generalized directly to random reward under infinite-horizon case, as long as the reward function is bounded. As a matter of fact, in our simulation, we set reward functions under both settings drawn from random normal distribution. Here we give the posterior distribution updating principle for transition probability distribution with the Dirichlet prior and also the random normal reward functions with random normal prior for the mean parameter.

**Posterior Distribution Update:**  For each pair of state-action $s, a$, the transition probability $P_{s,a}(i), i \in \mathcal{S}$ is a categorical distribution over state space $\mathcal{S}$. Dirichlet distribution is a conjugate prior for the categorical distribution which has the following property: given a prior Dirichlet$(\alpha_1, \alpha_2, \ldots, \alpha_S)$ for $P_{s,a}$, after observing a transition from $s$ to $i$ when taking action $a$, then the posterior distribution of $P_{s,a}$ becomes Dirichlet$(\alpha_1, \ldots, \alpha_i + 1, \ldots, \alpha_S)$. Hence when we update posterior distribution of transition probability $P_{s,a}$ after time $t$ (and before time $t + 1$), the posterior is Dirichlet$(\{N_{s,a}^t(s') + 1\}_{s' \in \mathcal{S}})$, where with a little bit abuse of notation, $N_{s,a}^t(s')$ denotes the number of times that $s$ jumps to $s'$ under action $a$ by (including) time $t$. For reward posterior, since we model the reward of each $(s, a)$ as $r \sim \mathcal{N}(\mu, 1)$ and start from a prior $\mu \sim \mathcal{N}(0, 1)$, after observing $n$ i.i.d. samples from $\mathcal{N}(\mu, 1)$, the posterior distribution for parameter $\mu$ is $\mathcal{N}(\hat{\mu}_n, 1/(n + 1))$, where $\hat{\mu}_n = \frac{1}{n} \sum_{i=1}^n r_i$ is the sample average of the observed rewards, and note that $\hat{\mu}_0$ is just 0.

---

**Algorithm 3:** Concurrent PSRL

**Data:** prior distribution $\phi, \mathcal{H}_0^p = \emptyset, \forall p \in [n]$
**for** *episode* $k = 1, 2, \ldots$ **do**
    **for** $p = 1 : n$ **do**
        Sample MDP $M_{kp} \sim \phi(\cdot | \mathcal{H}_{(k-1)H})$;
        Compute $\mu_{kp} \in \arg\max_\mu V_{\mu,1}^{M_{kp}}$;
        **for** *period h=1,...,H* **do**
            take action $a_{kh}^p = \mu_{kp}(s_{kh}^p, h)$;
            update $\mathcal{H}_{(k-1)H+h}^p = \mathcal{H}_{(k-1)H+h-1}^p \cup \{s_{kh}^p, a_{kh}^p, r_{kh}^p, s_{k(h+1)}^p\}$
        **end**
    **end**
    $\mathcal{H}_{kH} = \cup_{p=1}^n \mathcal{H}_{kH}^p$;  Update posterior distribution $\phi(\cdot | \mathcal{H}_{kH})$;
**end**

---

**Algorithm 4:** Concurrent Infinite-Horizon Posterior Sampling MDP under Dirichlet Prior

---

**Data:** State space $\mathcal{S}$, Action space $\mathcal{A}$, starting state vector across all agents $s_1 \in \mathcal{S}^P$, reward function $r$, time horizon $T$, $\mathcal{F}_0^p = \emptyset$ for $p \in [n]$.

**for** *epochs* $k = 1, 2, \ldots$ **do**

    Sample transition probability vectors: For each $s, a$ generate sample probability vectors $Q_{s,a}^k$ as follows:

    **Posterior sampling**: use samples from the posterior sampling:

$$Q_{s,a}^k \sim \phi(\cdot | \mathcal{F}_{k-1})$$

    Compute policy $\tilde{\pi}^k$: as the optimal gain policy for the sampled MDP $\tilde{M}^k$ constructed using sample set $\{Q_{s,a}^k, s \in \mathcal{S}, a \in \mathcal{A}\}$.

    Execute policy $\tilde{\pi}^k$:

    **for** *time steps* $t = \tau_k, \tau_k + 1, \ldots$, *until **break epoch*** **do**

        **for** *agent* $p = 1, 2, \ldots, n$ **do**

            (This block can be implemented in parallel)

            Play action $a_t^p = \tilde{\pi}_k(s_t^p)$.

            Observe the transition to the next state $s_{t+1}^p$.

            Update $\mathcal{F}_k^p = \mathcal{F}_k^p \cup \{s_t^p, a_t^p, r_t^p, s_{t+1}^p\}$

        **end**

        If $N_{s_t^p, a_t^p}^{t+1} \geq 2 N_{s_t^p, a_t^p}^{\tau_k}$ for some p, (i.e. for some $(s_t^p, a_t^p)$), then set $\tau_{k+1} = t + 1$ and **break epoch**.

        $\mathcal{F}_k = \cup_{p=1}^n \mathcal{F}_k^p$    Update posterior distribution $\phi(\cdot | \mathcal{F}_k)$;

    **end**

**end**

---

# B    Worst-case Bound of Infinite-Horizon Case under Dirichlet Prior

In this section, we provide a worst-case bound on the concurrent infinite-horizon posterior sampling method under Dirichlet Prior based on [3].

**Theorem B.1. (Worst-case Infinite-Horizon Regret Bound Under Dirichlet Prior)** With high probability, Algorithm 5 is bounded by

$$\mathcal{R}(T, \mathcal{M}, n) \leq \tilde{O}\left(DS\sqrt{ATn}\right)$$

*Proof.* As defined before,

$$\mathcal{R}(T, \mathcal{M}, n) = Tn\lambda^* - \sum_{p=1}^n \sum_{t=1}^T r(s_t^p, a_t^p)$$

$\lambda^*$ is the optimal gain of MDP $\mathcal{M}$. Algorithm 5 proceeds in epochs $k = 1, 2, \ldots, K$, where $K \leq SA \log(T)$. Now we define

$$R_k = (\tau_{k+1} - \tau_k)n\lambda^* - \sum_{t=\tau_k}^{\tau_{k+1}-1} \sum_{p=1}^n r_{s_t^p, a_t^p}$$

So we can write $R_k$ as

$$R_k = \sum_{t=\tau_k}^{\tau_{k+1}-1} \sum_{p=1}^n [\lambda^* - r_{s_t^p, a_t^p}]$$

$$= \sum_{t=\tau_k}^{\tau_{k+1}-1} \sum_{p=1}^n [(\lambda^* - \tilde{\lambda}_k) + (\tilde{\lambda}_k - r_{s_t^p, a_t^p})]$$

**Algorithm 5:** Concurrent Infinite-Horizon Posterior Sampling MDP under Dirichlet Prior

**Data:** State space $\mathcal{S}$, Action space $\mathcal{A}$, starting state vector across all agents $s_1 \in \mathcal{S}^n$, reward function $r$, time horizon $T$, parameters $\rho \in (0,1]$, $\psi$, $\omega$, $\kappa$, $\eta$.

**Initialize:** $\tau_1 = 1, \mathbf{M}_{s,a}^{\tau_1} = \omega\mathbf{1}$.

**for** *epochs* $k = 1, 2, \ldots$ **do**

    Sample transition probability vectors: For each $s, a$ generate $\psi$ independent sample probability vectors $Q_{s,a}^{j,k}, j = 1, 2, \ldots, \psi$ as follows:

        • (**Posterior sampling**): For $s, a$ such that $N_{s,a}^{\tau_k} \geq \eta$, use samples from the Dirichlet distribution:
$$Q_{s,a}^{j,k} \sim \text{Dirichlet}(\mathbf{M}_{s,a}^{\tau_k})$$

        • (**Simple optimistic sampling**): For $s, a$ such that $N_{s,a}^{\tau_k} < \eta$, use the following simple optimistic sampling: let
$$P_{s,a}^- = \hat{P}_{s,a} - \Delta$$
        where $\hat{P}_{s,a}(i) = \frac{N_{s,a}^{\tau_k}(i)}{N_{s,a}^{\tau_k}}$, and $\Delta_i = \min\{\sqrt{\frac{3\hat{P}_{s,a}(i)\log(4S)}{N_{s,a}^{\tau_k}}} + \frac{3\log(4S)}{N_{s,a}^{\tau_k}}, \hat{P}_{s,a}(i)\}$, and let $\mathbf{z}$ be a random vector picked uniformly at random from $\{\mathbf{1}_1, \ldots, \mathbf{1}_S\}$; set
$$Q_{s,a}^{j,k} = P_{s,a}^- + (1 - \sum_{i=1}^{S} P_{s,a}^-(i))\mathbf{z}$$

    Compute policy $\tilde{\pi}^k$: as the optimal gain policy for extended MDP $\tilde{M}^k$ constructed using sample set $\{Q_{s,a}^{j,k}, j = 1, 2, \ldots, \psi, s \in \mathcal{S}, a \in \mathcal{A}\}$.

    Execute policy $\tilde{\pi}^k$:

    **for** *time steps* $t = \tau_k, \tau_k + 1, \ldots$, *until* **break epoch** **do**

        **for** *agent* $p = 1, 2, \ldots, n$ **do**

            (This block can be implemented in parallel)

            Play action $a_t^p = \tilde{\pi}_k(s_t^p)$.

            Observe the transition to the next state $s_{t+1}^p$.

        **end**

        Set $N_{s,a}^{t+1}(i), M_{s,a}^{t+1}(i)$ for all $a \in \mathcal{A}, s, i \in \mathcal{S}$ as defined.

        If $N_{s_t^p, a_t^p}^{t+1} \geq 2N_{s_t^p, a_t^p}^{\tau_k}$ for some $p$, then set $\tau_{k+1} = t + 1$ and **break epoch**.

    **end**

**end**

where $\tilde{\lambda}_k$ is the optimal gain of the extended MDP $\tilde{M}^k$. From Lemma (2.1) we know that for any state $s$, optimal policy $\tilde{\pi}_k$ for communicating MDP $\tilde{M}^k$, action $a = \tilde{\pi}_k(s)$, $\lambda_k = r_{s,a} + \tilde{P}_{s,a}^T\tilde{h} - \tilde{h}_s$, where $\tilde{P}_{s,a} = Q_{s,a}^{j,k}$ for some $j$. So we have

$$\sum_{t=\tau_k}^{\tau_{k+1}-1} \sum_{p=1}^{n} (\tilde{\lambda}_k - r_{s_t^p, a_t^p}) = \sum_{t=\tau_k}^{\tau_{k+1}-1} \sum_{p=1}^{n} (\tilde{P}_{s_t^p, a_t^p} - P_{s_t^p, a_t^p} + P_{s_t^p, a_t^p} - \mathbf{1}_{s_t^p})^T\tilde{h}$$

$$= \sum_{t=\tau_k}^{\tau_{k+1}-1} \sum_{p=1}^{n} (\tilde{P}_{s_t^p, a_t^p} - P_{s_t^p, a_t^p})^T\tilde{h} + \sum_{t=\tau_k}^{\tau_{k+1}-1} \sum_{p=1}^{n} (P_{s_t^p, a_t^p} - \mathbf{1}_{s_t^p})^T\tilde{h}$$

$\square$

Similar to [3] we have the following result:

**Lemma B.1.** Fix any vector $h \in \mathbb{R}^S$ such that $|h_i - h_{i'}| \leq D$ for any $i, i' \in S$, and any epoch $k$. Let the parameters in Algorithm 5 take the following values:

$$\eta = \sqrt{\frac{nTS}{A} + 12\omega S^4}, \kappa = 120 \log(N_{s,a}^{\tau_k}/\rho), \omega = 720 \log(N_{s,a}^{\tau_k}/\rho)$$

Let $\Phi$ be the normal cumulative distribution function, then

$$\psi = Cn \log(nA/\rho), C \geq 7^{32/\phi}, \phi = \left(\frac{(1-\Phi)(1/2)^4}{2}\right)^4$$

Then for every $s, a$, with probability $1 - \frac{\rho}{SA}$ there exists at least one $j$ such that

$$(Q_{s,a}^{j,k})^T h \geq P_{s,a}^T h - O\left(D \log^2(Tn/\rho)\sqrt{\frac{SA}{Tn}}\right)$$

*Proof.* When $N_{s,a}^{\tau_k} > \eta$, we know that $Q_{s,a}^{j,k} \sim \text{Dirichlet}(m\bar{p}_1, \ldots, m\bar{p}_S)$, where $m = \frac{N_{s,a}^{\tau_k} + \omega S}{\kappa}$, $\bar{p}_i = \frac{N_{s,a}^{\tau_k}(i) + \omega}{N_{s,a}^{\tau_k} + \omega S}$. From [3] we know that with probability $\Omega(1/S) - 8S\rho$,

$$(Q_{s,a}^{j,k} - P_{s,a})^T h \geq -O\left(\omega \frac{DS \log(N_{s,a}^{\tau_k}/\rho)}{N_{s,a}^{\tau_k}}\right) \geq -O\left(D \log^2(Tn/\rho)\sqrt{\frac{SA}{nT}}\right)$$

When $n < \eta$, with probability at least $1/(2S)$, $(Q_{s,a}^{j,k} - P_{s,a})^T h \geq 0$. So we have that with probability $\Omega(1/S - S\rho)$,

$$(Q_{s,a}^{j,k})^T h \geq P_{s,a}^T h - O\left(D \log^2(Tn/\rho)\sqrt{\frac{SA}{Tn}}\right)$$

for at least one sample $j$. And the rest of the proof follows similarly as in [3] Lemma 5.3. $\qquad \square$

Similar to [3], the above Lemma directly leads to the following result:

**Lemma B.2.** With probability $1 - \rho$, for every epoch $k$, the optimal gain $\tilde{\lambda}_k$ of the extended MDP $\tilde{M}^k$ satisfies:

$$\tilde{\lambda}_k \geq \lambda^* - O\left(D \log^2(Tn/\rho)\sqrt{\frac{SA}{nT}}\right)$$

where $\lambda^*$ is the optimal gain of MDP $\mathcal{M}$ and $D$ is the diameter.

*Proof.* The proof follows in the same way as in Lemma 5.4 of [3]. $\qquad \square$

**Lemma B.3.** (Bound on number of epochs). When $Tn \geq SA$, the number of epochs $K$ of Algorithm 5 is upper bounded by

$$K \leq SA \log_2\left(\frac{8Tn}{SA}\right)$$

*Proof.* We use the technique similar to [4] to give the upper bound for the number of epochs. Define $N(s,a) = \{t < T+1 : s_t = s, a_t = a\}$ be the total number of observations of the state-action pair $(s,a)$ up to time step $T$. Let $v_k(s,a)$ be the number of observations of state-action pair $(s,a)$ within epoch $k$. For each epoch $k < K$ there exists a state-action pair $(s,a)$ with $v_k(s,a) = N_{s,a}^{\tau_k}$, or $v_k(s,a) = 1, N_{s,a}^{\tau_k} = 0$. Let $K(s,a)$ be the number of epochs with $v_k(s,a) = N_{s,a}^{\tau_k}$ and $N_{s,a}^{\tau_k} > 0$. If $N(s,a) > 0$, then $v_k(s,a) = N_{s,a}^{\tau_k}$ implies $N_{s,a}^{\tau_{k+1}} = 2N_{s,a}^{\tau_k}$. so that

$$N(s,a) = \sum_{k=1}^{K} v_k(s,a) \geq 1 + \sum_{k:v_k(s,a)=N_{s,a}^{\tau_k}} N_{s,a}^{\tau_k} \geq 1 + \sum_{i=1}^{K(s,a)} 2^{i-1} = 2^{K(s,a)}$$

If $N(s, a) = 0$, then $K(s, a) = 0$, hence $N(s, a) \geq 2^{K(s,a)} - 1$ for any state-action pair $(s, a)$. It follows that

$$Tn = \sum_{s,a} N(s, a) \geq \sum_{s,a} (2^{K(s,a)} - 1) \tag{B.1}$$

In each epoch $k$, note that there exists a state-action pair $(s, a)$, such that it's visited with either $N_{s,a}^{\tau_k} = 0$ or $N_{s,a}^{\tau_k} = v_k(s, a)$. So we have

$$K \leq 1 + SA + \sum_{s,a} K(s, a) \Rightarrow \sum_{s,a} K(s, a) \geq K - 1 - SA$$

This implies

$$\sum_{s,a} 2^{K(s,a)} \geq SA2^{\sum_{s,a} K(s,a)/SA} \geq SA2^{\frac{K-1}{SA} - 1}$$

With (F.1) we have:

$$Tn \geq SA(2^{\frac{K-1}{SA} - 1} - 1)$$

So we have

$$K \leq 1 + 2SA + SA \log_2(\frac{Tn}{SA})$$

So when $Tn \geq SA$, we have

$$K \leq SA \log_2(\frac{8Tn}{SA})$$

$\square$

**Lemma B.4.** ([3] Lemma 5.5). In every epoch $k$, with probability $1 - \rho$, for all samples $j$, all $s, a$, and all vectors $h \in [0, H]^S$,

$$(Q_{s,a}^{j,k} - P_{s,a})^T h \leq O\Big(H(\sqrt{\frac{S}{N_{s,a}^{\tau_k}}} + \frac{S}{N_{s,a}^{\tau_k}}) \log^2(SAT/\rho)\Big)$$

**Lemma B.5.** (Diameter of the extended MDP). Assume $Tn > CSA \log^4(SATn/\rho)$ for some constant $C$ sufficiently large. The bias vector of the extended MDP $\tilde{M}^k$ satisfies

$$\max_s \tilde{h}_s - \min_s \tilde{h}_s \leq D(\tilde{M}^k) \leq 2D$$

with probability $1 - \rho$.

*Proof.* The proof follows that of Lemma 5.7 in [3]. $\square$

**Lemma B.6.** With probability $1 - \rho$,

$$\sum_{k=1}^{K} \sum_{p=1}^{n} (Q_{s_t^p,a_t^p}^{j,k} - P_{s_t^p,a_t^p})^T \tilde{h} \leq O\Big(DS\sqrt{AT}(\log(SA) + \log\log_2(\frac{8Tn}{SA})) \log^2(\frac{SAT}{\rho})$$

$$+ DS^3 A^2 \log_2(\frac{8Tn}{SA}) \log^2(\frac{SAT}{\rho})\Big)$$

*Proof.* From Lemma B.4, and note the fact that $N_{s,a}^{\tau_{k+1}} - N_{s,a}^{\tau_k} \leq N_{s,a}^{\tau_k}$ for any pair of $(s,a)$, we have

$$\sum_{p=1}^{n}(Q_{s_t^p,a_t^p}^{j,k} - P_{s_t^p,a_t^p})^T\tilde{h} \leq D\sum_{s,a}(N_{s,a}^{\tau_{k+1}} - N_{s,a}^{\tau_k})(\frac{\sqrt{S}}{\sqrt{N_{s,a}^{\tau_k}}} + \frac{S}{N_{s,a}^{\tau_k}})\log^2(\frac{SAT}{\rho})$$

$$\leq O\Big((D\sqrt{S}\sum_{s,a}\sqrt{N_{s,a}^{\tau_k}} + DS^2A)\log^2(\frac{SAT}{\rho})\Big)$$

So if we use the fact that $N_{s,a}^{\tau_{k+1}} \leq 2N_{s,a}^{\tau_k}$, we have

$$\sum_{k=1}^{K}\sum_{p=1}^{n}(Q_{s_t^p,a_t^p}^{j,k} - P_{s_t^p,a_t^p})^T\tilde{h} \leq O\Big(\sum_{k=1}^{K}(D\sqrt{S}\sum_{s,a}\sqrt{N_{s,a}^{\tau_k}} + DS^2A)\log^2(\frac{SAT}{\rho})\Big)$$

$$\leq O\Big((D\sqrt{S}\log(K)\sum_{s,a}\sqrt{N_{s,a}^{\tau_K}} + KDS^2A)\log^2(\frac{SAT}{\rho})\Big)$$

Also by simple worst case analysis, using the fact that $\sum_{s,a}N_{s,a}^{\tau_K} \leq T$ we have that. with probability at least $1-\rho$,

$$\sum_{s,a}\sqrt{N_{s,a}^{\tau_K}} \leq \sqrt{SAT}$$

So using Lemma F.1 we have that with probability at least $1-\rho$

$$\sum_{k=1}^{K}\sum_{p=1}^{n}(Q_{s_t^p,a_t^p}^{j,k} - P_{s_t^p,a_t^p})^T\tilde{h} \leq O\Big((D\sqrt{S}\log(K)\sum_{s,a}\sqrt{N_{s,a}^{\tau_K}} + KDS^2A)\log^2(\frac{SAT}{\rho})\Big)$$

$$\leq O\Big(DS\sqrt{AT}(\log(SA) + \log\log_2(\frac{8TP}{SA}))\log^2(\frac{SAT}{\rho})$$

$$+ DS^3A^2\log_2(\frac{8TP}{SA})\log^2(\frac{SAT}{\rho})\Big)$$

$\square$

**Lemma B.7.** With probability at least $1-2\delta$

$$\sum_{t=\tau_k}^{\tau_{k+1}}\sum_{p=1}^{n}(P_{s_t^p,a_t^p} - \mathbf{1}_{s_t^p})^T\tilde{h} \leq O\Big(D\sqrt{n(\tau_{k+1} - \tau_k)\log(n/\delta)}\Big) \tag{B.2}$$

*Proof.* We expand the left hand side of (G.2) as

$$\sum_{t=\tau_k}^{\tau_{k+1}-1}\sum_{p=1}^{n}(P_{s_t^p,a_t^p} - \mathbf{1}_{s_t^p})^T\tilde{h} = \sum_{t=\tau_k}^{\tau_{k+1}-1}\sum_{p=1}^{n}[P_{s_t^p,a_t^p} - \mathbf{1}_{s_{t+1}^p} + \mathbf{1}_{s_{t+1}^p} - \mathbf{1}_{s_t^p}]^T\tilde{h}$$

$$= \underbrace{\sum_{t=\tau_k}^{\tau_{k+1}-1}\sum_{p=1}^{n}(P_{s_t^p,a_t^p} - \mathbf{1}_{s_{t+1}^p})^T\tilde{h}}_{(1)} + \underbrace{\sum_{t=\tau_k}^{\tau_{k+1}-1}\sum_{p=1}^{n}(\mathbf{1}_{s_{t+1}^p} - \mathbf{1}_{s_t^p})^T\tilde{h}}_{(2)}$$

**Bound on part (1):**

Let $Z_p = \sum_{t=\tau_k}^{\tau_{k+1}-1}(P_{s_t^p,a_t^p} - \mathbf{1}_{s_{t+1}^p})^T\tilde{h}$ for each $p$. Let $\mathcal{E}_k$ be the event defined as

$$\mathcal{E}_\delta^k = \{|Z_p| \le D\sqrt{8(\tau_{k+1}-\tau_k)}\log(4n/\delta), \forall p\}$$

Let

$$A_\delta^k = \{\sum_{t=\tau_k}^{\tau_{k+1}-1}\sum_{p=1}^{n}(P_{s_t^p,a_t^p} - \mathbf{1}_{s_{t+1}^p})^T\tilde{h} \ge 4D\sqrt{n(\tau_{k+1}-\tau_k)}\log(\frac{\sqrt{8n}}{\delta})\}$$

Thus

$$\mathbb{P}[\sum_{t=\tau_k}^{\tau_{k+1}-1}\sum_{p=1}^{n}(P_{s_t^p,a_t^p} - \mathbf{1}_{s_{t+1}^p})^T\tilde{h} \ge 4D\sqrt{n(\tau_{k+1}-\tau_k)}\log(\frac{\sqrt{8n}}{\delta})] = \mathbb{P}(A_\delta^k, \mathcal{E}_\delta^k) + \mathbb{P}(A_\delta^k, (\mathcal{E}_\delta^k)^c)$$

$$\le \mathbb{P}(A_\delta^k, \mathcal{E}_\delta^k) + \mathbb{P}((\mathcal{E}_\delta^k)^c)$$

Note that for any $p, t$,

$$\mathbb{E}[\mathbf{1}_{s_{t+1}^p}^T\tilde{h}|\tilde{\pi}_k, \tilde{h}, s_t^p] = P_{s_t^p,a_t^p}^T\tilde{h} \tag{B.3}$$

On event $\mathcal{E}_\delta^k$ we have that

$$\mathbb{P}(A_\delta^k, \mathcal{E}_\delta^k) = \mathbb{P}(A_\delta^k|\mathcal{E}_\delta^k)\mathbb{P}(\mathcal{E}_\delta^k) \le \mathbb{P}(A_\delta^k|\mathcal{E}_\delta^k)$$

Note that by the fact that $\sqrt{ab} \le \frac{a+b}{2}$ for $a, b > 0$, we have

$$\sqrt{\log(\frac{2}{\delta})\log(\frac{4n}{\delta})} \le \frac{1}{2}(\log(\frac{2}{\delta}) + \log(\frac{4n}{\delta})) = \log(\frac{\sqrt{8n}}{\delta})$$

By (F.5), $(P_{s_t^p,a_t^p} - \mathbf{1}_{s_{t+1}^p})^T\tilde{h}$ is a martingale difference sequence for each $p$. So by Hoeffding-Azuma inequality,

$$\mathbb{P}(A_\delta^k|\mathcal{E}_\delta^k) = \mathbb{P}(\sum_{t=\tau_k}^{\tau_{k+1}-1}\sum_{p=1}^{n}(P_{s_t^p,a_t^p} - \mathbf{1}_{s_{t+1}^p})^T\tilde{h} \ge 4D\sqrt{n(\tau_{k+1}-\tau_k)}\log(\frac{\sqrt{8n}}{\delta}))$$

$$\le \mathbb{P}(\sum_{t=\tau_k}^{\tau_{k+1}-1}\sum_{p=1}^{n}(P_{s_t^p,a_t^p} - \mathbf{1}_{s_{t+1}^p})^T\tilde{h} \ge 4D\sqrt{n(\tau_{k+1}-\tau_k)}\sqrt{\log(\frac{2}{\delta})\log(\frac{4n}{\delta})})$$

$$\le \frac{\delta}{2}$$

Again by Hoeffding-Azuma inequality and the fact that the diameter of the extended MDP is bounded by $2D$, $Z_p$ are i.i.d. across $p$, we have

$$\mathbb{P}((\mathcal{E}_\delta^k)^c) \le \sum_{p=1}^{P}\mathbb{P}(|Z_p| \ge D\sqrt{8(\tau_{k+1}-\tau_k)\log(4P/\delta)}) \le 2P\frac{\delta}{4P} = \frac{\delta}{2}$$

Hence

$$\mathbb{P}(A_\delta^k) \le \mathbb{P}(A_\delta^k|\mathcal{E}_\delta^k) + \mathbb{P}((\mathcal{E}_\delta^k)^c) \le \frac{\delta}{2} + \frac{\delta}{2} = \delta$$

**Bound on part (2):**

$$(2) = \sum_{t=\tau_k}^{\tau_{k+1}-1} \sum_{p=1}^{n} (\mathbf{1}_{s_{t+1}^p} - \mathbf{1}_{s_t^p})^T \tilde{h}$$

$$= \sum_{p=1}^{n} (\mathbf{1}_{s_{\tau_{k+1}}^p} - \mathbf{1}_{s_{\tau_k}^p})^T \tilde{h}$$

Note that $(\mathbf{1}_{s_{\tau_{k+1}}^p} - \mathbf{1}_{s_{\tau_k}^p})^T \tilde{h}$ are. i.i.d. across $p = 1, 2, \ldots, n$, and that

$$|(\mathbf{1}_{s_{\tau_{k+1}}^p} - \mathbf{1}_{s_{\tau_k}^p})^T \tilde{h}| \le 2D$$

Hence using Azuma-Hoeffding inequality, we have that for all $\epsilon > 0$,

$$\mathbb{P}(\sum_{p=1}^{n} (\mathbf{1}_{s_{\tau_{k+1}}^p} - \mathbf{1}_{s_{\tau_k}^p})^T \tilde{h} \ge \epsilon) \le \exp(-\frac{\epsilon^2}{8nD^2})$$

Taking $\epsilon = D\sqrt{8n \log(1/\delta)}$ in the above, then we have that with probability at least $1 - \delta$,

$$\sum_{p=1}^{n} (\mathbf{1}_{s_{\tau_{k+1}}^p} - \mathbf{1}_{s_{\tau_k}^p})^T \tilde{h} \le D\sqrt{8n \log(1/\delta)} \qquad (B.4)$$

$\square$

So from Lemma (B.2), (F.2), (G.2), we have the following bound on the regret of Algorithm 5.

**Theorem B.2.** With probability at least $1 - 5\rho$,

$$\mathcal{R}(T, \mathcal{M}, n) \le O\Big(D\sqrt{SA}\log^2(Tn/\rho) + DS\sqrt{AT}\log^3(\frac{SAT}{\rho}) + DS^3A^2\log(\frac{SAT}{\rho})\log(\frac{Tn}{SA}) + D\sqrt{nT\log(n/\rho)}\Big)$$

This indicates that With high probability

$$\mathcal{R}(T, \mathcal{M}, n) \le \tilde{O}\Big(DS\sqrt{ATn}\Big)$$

*Proof.* Note that

$$\sum_k \tau_{k+1} - \tau_k \le T \quad \text{and} \quad \sum_{k=1}^{K} \sqrt{\tau_{k+1} - \tau_k} \le \sqrt{KT}$$

From Lemma (B.2), (F.2), (G.2), we have that with probability at least $1 - 5\rho$,

$$\mathcal{R}(T, \mathcal{M}, n) \le O\Big(D\sqrt{SA}\log^2(Tn/\rho) + DS\sqrt{AT}\log^3(\frac{SAT}{\rho}) + DS^3A^2\log(\frac{SAT}{\rho})\log(\frac{Tn}{SA}) + D\sqrt{nT\log(n/\rho)}\Big)$$

So we have the bound as indicated above. $\square$

# C   Simulation Results

In both finite and infinite horizon cases, we use an MDP with transition probabilities drawn from Dirichlet$(1, 1, 1, \ldots, 1)$ and deterministic reward functions drawn from $\mathcal{N}(0, 1)$. When we model the posterior of the MDP, we start from a Dirichlet$(1, 1, 1, \ldots, 1)$ as the prior distribution of the transition probabilities, and a $\mathcal{N}(0, 1)$ as the prior distribution of the mean ($\mu$) of the i.i.d. Gaussian rewards $r \sim \mathcal{N}(\mu, 1)$. For infinite horizon case, the epochs are defined using doubling epoch strategy as defined in Algorithm (2).

Under both settings, the policy at each episode or epoch is computed using value iteration from transition probabilities sampled from a Dirichlet distribution with parameters being the number of visits to each of the state, action, and next states tuples. The discount for value iteration is set to 0.99 and tolerance 0.01. All experiments were run using CPUs, either through a desktop computer or Google Cloud Platform. Due to page limitation, we put the experiment results in the appendix.

**Hyperparameters and Results:** For finite horizon case, we run our experiments with different settings as in the Table 1. We sample 10 environments and average over their per-agent total regrets to approximate the Bayesian per-agent regret. For infinite horizon case, we run our experiments with the settings as in Table 2. Again, we sample 10 environments and average over their per-agent total regrets to approximate the Bayesian per-agent regret. We plot the Bayesian per-agent total regrets against the number of agents for both finite-horizon and infinite-horizon settings. Figure 1 shows finite-horizon results under two different $K$ and $H$ settings, i.e., $K = 30, H = 75$ and $K = 20, H = 10$ respectively. The $\Theta(\frac{1}{\sqrt{n}})$ trends are plotted as dashed curves for reference. We can see that under all 4 settings, the Bayesian per-agent regrets from simulations fit well the Bayesian regret bounds given by our theory. Figure 4 shows infinite-horizon results under two different settings of total time steps, i.e., $T = 1000$ and $T = 2000$ respectively. Again, the simulated Bayesian per-agent regret fits well the $\Theta(\frac{1}{\sqrt{n}})$ trend given by our Bayesian regret bounds.

Table 1: Finite-Horizon Settings

| S (state space size) | A (action space size) | K (number of episodes) | H (horizon) |
|---|---|---|---|
| 5 | 5 | 20 | 10 |
| 5 | 5 | 30 | 75 |
| 20 | 10 | 20 | 10 |
| 20 | 10 | 30 | 75 |

Table 2: Infinite-Horizon Settings

| S (state space size) | A (action space size) | T (total time steps for running) |
|---|---|---|
| 10 | 5 | 1000 |
| 10 | 5 | 2000 |

**Concurrent UCB Comparison:** We compare our concurrent PSRL to a baseline concurrent UCB algorithm to verify its effectiveness. For concurrent UCB, we parallelize UCB action selection on a TD-based reinforcement learning algorithm by syncing the experience of all agents at the end of each episode similar to concurrent PSRL. For the infinite horizon case, we again split training based on epochs and break the current epoch when the number of times an action is selected for some action is greater than two times in the previous epoch. For the finite horizon case, we ran experiments for all settings in Table 1, and for the infinite horizon case we ran experiments for all settings in Table 2 to compare our concurrent PSRL to concurrent UCB. The results show that across all settings for both finite-horizon (Figure 4a, 4b, 3c, 3d) and infinite (Figure 4) cases, concurrent PSRL outperforms concurrent UCB in terms of Bayesian regret.

# D  Proof of Finite-Horizon Case under Dirichlet Prior

**Decomposition of the Regret:** Note that we have
$$\text{BayesRegret}(T, \pi, \phi) = \mathbb{E}[\text{Regret}(T, \pi, M^*)|M^* \sim \phi]$$
$$= \sum_{k=1}^{\lceil T/H \rceil} \sum_{p=1}^{n} \mathbb{E}[\sum_{s \in \mathcal{S}} \rho(s)(V_{\pi^*,1}^{M^*}(s) - V_{\pi_{kp},1}^{M^*}(s))|M^* \sim \phi]$$
And we can write
$$V_{\pi^*,1}^{M^*}(s) - V_{\pi_{kp},1}^{M^*(s)} = \underbrace{V_{\pi^*,1}^{M^*}(s) - V_{\pi_{kp},1}^{M_{kp}}(s)}_{\Delta_{kp}^{\text{opt}}} + \underbrace{V_{\pi_{kp},1}^{M_{kp}}(s) - V_{\pi_{kp},1}^{M^*}(s)}_{\Delta_{kp}^{\text{conc}}}$$

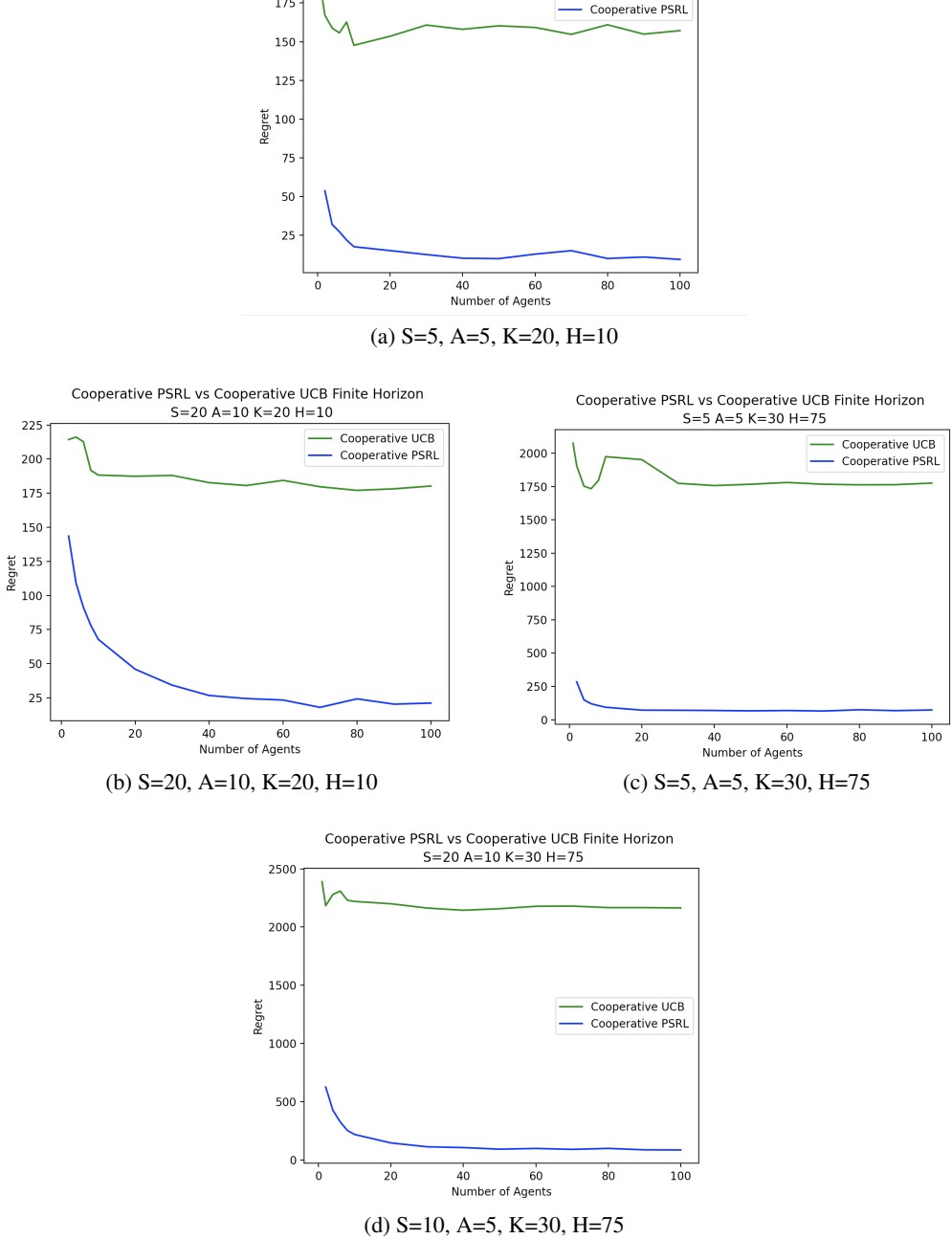

(a) S=5, A=5, K=20, H=10

(b) S=20, A=10, K=20, H=10

(c) S=5, A=5, K=30, H=75

(d) S=10, A=5, K=30, H=75

Figure 3: Bayesian regrets for Finite-Horizon Concurrent TS versus Concurrent UCB method with Dirichlet Prior for Transition Probabilities and Gaussian prior for rewards.

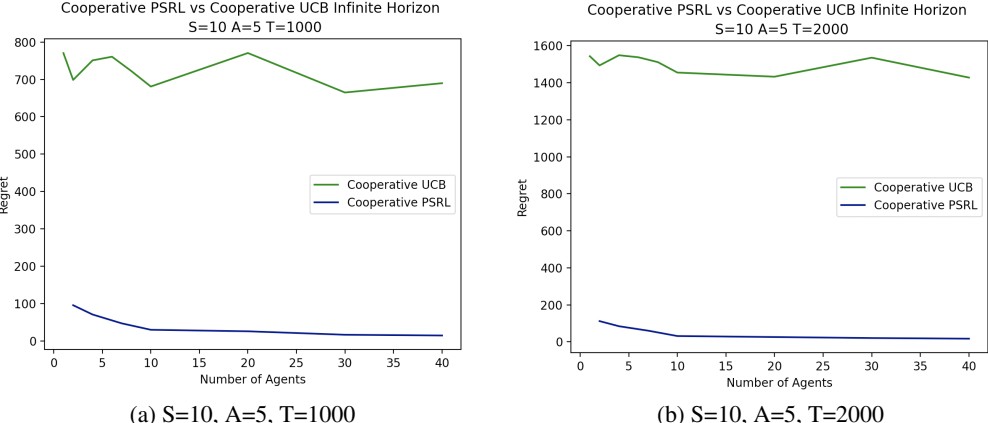

(a) S=10, A=5, T=1000            (b) S=10, A=5, T=2000

Figure 4: Bayesian regrets for Infinite-Horizon Concurrent TS versus Concurrent UCB method with Dirichlet Prior for Transition Probabilities and Gaussian prior for rewards.

Note that conditioned upon any data $\mathcal{H}_{(k-1)H}$, the true MDP $M^*$ and the sampled $M_{kp}$ are identically distributed. This means that $\mathbb{E}[\Delta_{kp}^{\text{opt}}] \leq 0$ for all $k, p$. Therefore, we just need to bound $\sum_{k=1}^{\lceil T/H \rceil} \sum_{p=1}^{P} \mathbb{E}[\Delta_{kp}^{\text{conc}} | \mathcal{H}_{(k-1)H}]$. For the convenience of notation, we write $V_{kh}^{kp} = V_{\pi_{kp},h}^{M_{k,p}}$. Now we rewrite the regret via the Bellman operator, with $w^R(x) = \bar{r}_{kp}(x) - \hat{r}_k(x), w_h^P(x) = (P_{kp}(x) - \hat{P}_k(x))^T V_{k,h+1}^{kp}$:

$$
\begin{aligned}
\mathbb{E}[\Delta_{kp}^{\text{conc}} | \mathcal{H}_{(k-1)H}] &= \mathbb{E}[(\bar{r}_{kp} - \bar{r}^*)(x_{k1}^p) + P_{kp}(x_{k1}^p)^T V_{k2}^{kp} - P^*(x_{k1}^p)V_{k2}^* | \mathcal{H}_{(k-1)H}] \\
&= \mathbb{E}[(\bar{r}_{kp} - \bar{r}^*)(x_{k1}^p) + (P_{kp}(x_{k1}^p) - \hat{P}_k(x_{k1}^p))^T V_{k2}^{kp} + \mathbb{E}[(V_{k2}^{kp} - V_{k2})(s') | s' \sim P^*(x_{k1}^p)] | \mathcal{H}_{(k-1)H}] \\
&= \ldots \\
&= \mathbb{E}[\sum_{h=1}^{H}(\bar{r}_{kp}(x_{k1}^p) - \hat{r}^*(x_{k1}^p)) + \sum_{h=1}^{H}\{(P_{kp}(x_{kh}^p) - \hat{P}_k(x_{kh}^p))V_{kh}^{kp}\} | \mathcal{H}_{(k-1)H}] \\
&\leq \mathbb{E}[\sum_{h=1}^{H}|w^R(x_{kh}^p)| + \sum_{h=1}^{H}|w_h^P(x_{k,h+1}^p)| | \mathcal{H}_{(k-1)H}]
\end{aligned}
$$

### D.1 Useful Lemmas

**Lemma D.1.** (Posterior Sampling [16]). For any $\mathcal{H}_{(k-1)H}$-measurable function $g$,
$$
\mathbb{E}[g(M^*) | \mathcal{H}_{(k-1)H}] = \mathbb{E}[g(M_k) | \mathcal{H}_{(k-1)H}]
$$

**Lemma D.2.** (Confidence Interval [4]) For any $t \geq 1$, the probability that the true MDP $M^*$ is not contained in the set of plausible MDPs $M_t$ at time $t$ (as given by the confidence intervals in the below) is at most $\frac{\delta}{15t^6}$, where $M_t$ is define to be the set of all MDPs with states and actions as in $M^*$, and with transition probabilities $\tilde{p}(\cdot|s, a)$ close to $\hat{p}_k(\cdot|s, a)$, and rewards $\tilde{r}(s, a) \in [0, 1]$ close to $\hat{r}_k(s, a)$:

$$
|\tilde{r}(s, a) - \hat{r}_k(s, a)| \leq \sqrt{\frac{7 \log(2SA\tau_k/\delta)}{2 \max\{1, N_k(s, a)\}}}, \quad \|\tilde{p}(\cdot|s, a) - \hat{p}_k(\cdot|s, a)\|_1 \leq \frac{14S \log(2A\tau_k/\delta)}{\max\{1, N_k(s, a)\}}
$$

where $\hat{p}_k$ is the empirical distribution by time $\tau_k$ and $\hat{r}_k$ is the observed average accumulated rewards, which will be given more clear definitions in the following.

**Lemma D.3.** (Empirical Distribution Deviation [31])The $L^1$-deviation of the true distribution and the empirical distribution over $m$ distinct events from $n$ samples is bounded by

$$
\mathbb{P}(\left\|\hat{P}(\cdot) - P(\cdot)\right\|_1 \geq \epsilon) \leq (2^m - 2)\exp(-\frac{n\epsilon^2}{2}) \tag{D.1}
$$

**Lemma D.4.** (Sub-Gaussian tail bounds [21]) Let $x_1, \ldots, x_n$ be independent samples from sub-Gaussian random variables. Then, for any $\delta > 0$,

$$\mathbb{P}(\frac{1}{n}|\sum_{i=1}^{n} x_i| \geq \sqrt{\frac{2\log(2/\delta)}{n}}) \leq \delta$$

**Lemma D.5.** (Gaussian-Dirichlet dominance [21]) For all fixed $V \in [0,1]^N$, $\alpha \in [0, \infty)^N$ with $\alpha^T \mathbf{1} \geq 2$, if $X \sim \mathcal{N}(\alpha^T V \alpha^T \mathbf{1}, 1/\alpha^T \mathbf{1})$ and $Y = P^T V$ for $P \sim \text{Dirichlet}(\alpha)$ then $X \succeq_{\text{so}} Y$.

**Lemma D.6.** (Transition Concentration [21]) For any independent prior over rewards with $r \in [0,1]$, additive sub-Gaussian noise and an independent Dirichlet prior over transitions at state-action pair $x_{kh}$, then

$$w_h^P(x_{kh}) \leq 2H\sqrt{\frac{2\log(2/\delta)}{\max(n_k(x_{kh}) - 2, 1)}}$$

with probability at least $1 - \delta$.

### D.2  Proof of Theorem (3.1)

*Proof.* Let $\hat{P}_a^t(\cdot|s)$ denote the empirical distribution up to period $t$ of transitions observed after sampling $(s, a)$, let $\hat{R}_a^t(s)$ denote the empirical average reward. Define $N_{t_k}(s, a) = \sum_{t=1}^{t_k-1} \mathbf{1}\{(s_t, a_t) = (s, a)\}$ to be the number of times $(s, a)$ was sampled prior to time $t_k$, where $t_k = (k-1)H + 1$ is defined to be the start time step of period $k$. Empirical estimate of rewards and transition probabilities at each $(s, a)$ are defined as

$$\hat{R}_a^{t_k}(s) = \frac{R_k(s, a)}{\max\{1, N_k(s, a)\}} \quad \hat{P}_a^{t_k}(s'|s) = \frac{P_k(s, a, s')}{\max\{1, N_k(s, a)\}}$$

where

$$R_k(s, a) = \sum_{\tau=1}^{t_k-1} r_\tau \mathbf{1}[s_\tau = s, a_\tau = a]; \quad P_k(s, a, s') = \#\{\tau < t_k : s_\tau = s, a_\tau = a, s_{\tau+1} = s'\}$$

Note that $N_{t_k}(s, a) = \sum_{t=1}^{t_k-1} \mathbf{1}\{(s_t, a_t) = (s, a)\}$ can take values in $\{0, 1, 2, \ldots, P(t_k - 1)\}$.

Define the confidence set for episode $k$:

$$\mathcal{M}_k = \{M : \left\|\hat{P}_a^{t_k}(\cdot|s) - P_a^M(\cdot|s)\right\|_1 \leq B_{k1}, \ |\hat{R}_a^{t_k}(s) - R_a^M(s)| \leq B_{k2} \ \forall(s, a)\} \quad \text{(D.2)}$$

where

$$B_{k1} = \sqrt{\frac{14S}{\max\{1, N_k(s_{kh}^p, \pi_{kp})\}} \log(\frac{2At_k}{\delta})}$$

$$B_{k2} = \sqrt{\frac{7}{2\max\{1, N(s_{kh}^p, \pi_{kp})\}} \log(\frac{2SAt_k}{\delta})}$$

From Lemma (D.1), we know that $\mathcal{M}_k$ is $\mathcal{H}_{(k-1)H}$-measurable, and that

$$\mathbb{E}[\mathbf{1}\{M_k \notin \mathcal{M}_k\}|\mathcal{H}_{(k-1)H}] = \mathbb{E}[\mathbf{1}\{M^* \notin \mathcal{M}_k\}|\mathcal{H}_{(k-1)H}] \quad \text{(D.3)}$$

We rewrite our previous bound here:

$$\mathbb{E}[\Delta_{kp}^{\text{conc}}|\mathcal{H}_{(k-1)H}] \leq \mathbb{E}[\sum_{h=1}^{H} |w^R(x_{kh}^p)| + \sum_{h=1}^{H} |w_h^P(x_{k,h+1}^p)||\mathcal{H}_{(k-1)H}] \quad \text{(D.4)}$$

where

$$w^R(x) = \bar{r}_{kp}(x) - \tilde{r}_k(x)$$

$$w_h^P(x) = (P_{kp}(x) - \tilde{P}_k(x))^T V_{k,h+1}^{kp}$$

$$\bar{r}_{kp}(x_{kh}^p) = \mathbb{E}[r | r \sim R^{M_{kp}}(s_{kh}^p, \pi_{kp})], \quad \tilde{r}_k(x_{kh}^p) = \mathbb{E}[\bar{r}^*(s_{kh}^p, \pi_{kp}) | \mathcal{H}_{(k-1)H}]$$

$$P_{kp}(s_{kh}^p, \pi_{kp}) = P_{\pi_{kp}}^{M_{kp}}(\cdot | s_{kh}^p), \quad \tilde{P}_k(s_{kh}^p, \pi_{kp}) = \mathbb{E}[P^*(s_{kh}^p, \pi_{kp}) | \mathcal{H}_{(k-1)H}]$$

For $\mathcal{E}_k^p = \{M^* \in \mathcal{M}_k, M_{kp} \in \mathcal{M}_k\}$, we will condition on event $\mathcal{E}_k^p$ and $(\mathcal{E}_k^p)^c$ to bound (D.4).
From Lemma (D.3), for $m = S, t = t_k$, let

$$\epsilon = \sqrt{\frac{2}{n} \log(\frac{2^S 20 S A t^7}{\delta})} \le \sqrt{\frac{14S}{n} \log(\frac{2At}{\delta})}$$

From Lemma (D.1),

$$\mathbb{P}\Big( \left\| P(\cdot | s, a) - \hat{P}^t(\cdot | s, a) \right\|_1 \ge \sqrt{\frac{14S}{N} \log(\frac{2At}{\delta})} \Big) \le 2^S \exp(-\frac{n}{2} \frac{2}{n} \log(\frac{2^S 20 S A t^7}{\delta}))$$

$$\le \frac{\delta}{20 t^7 S A}$$

where $\hat{P}$ is the empirical distribution, and $P$ is the true transition distribution.

$$\mathbb{P}\Big( \left| \hat{r}^t(s, a) - \bar{r}(s, a) \right| \ge \sqrt{\frac{7}{2N} \log(\frac{2SAt}{\delta})} \Big) \le \frac{\delta}{60 t^7 S A}$$

where $\hat{r}^t$ is the empirical mean reward, and $\bar{r}$ is the true mean reward. Hence a union bound over all possible values of $N = 1, 2, \ldots, n(t-1)$ gives

$$\mathbb{P}\Big( \left\| P(\cdot | s, a) - \hat{P}(\cdot | s, a) \right\|_1 \ge \sqrt{\frac{14S}{\max\{1, N(s, a)\}} \log(\frac{2At}{\delta})} \Big) \le \min\{ \sum_{n=1}^{n(t-1)} \frac{\delta}{20 t^7 S A}, 1 \} < \min\{\frac{n\delta}{20 t^6 S A}, 1\}$$

$$\mathbb{P}\Big( \left| \hat{r}^t(s, a) - \bar{r}(s, a) \right| \ge \sqrt{\frac{7}{2 \max\{1, N(s, a)\}} \log(\frac{2SAt}{\delta})} \Big) \le \min\{ \sum_{N=1}^{n(t-1)} \frac{\delta}{60 t^7 S A}, 1 \} < \min\{\frac{n\delta}{60 t^6 S A}, 1\}$$

So if we condition on the event that $\mathcal{E}_k^p = \{M^* \in \mathcal{M}_k, M_k^p \in \mathcal{M}_k\}$, we have

$$\mathbb{E}[\Delta_{kp}^{\text{conc}} | \mathcal{H}_{(k-1)H}, \mathcal{E}_k^p] \le \mathbb{E}[\sum_{h=1}^H |w^R(x_{kh}^p)| + \sum_{h=1}^H |w_h^P(x_{k,h+1}^p)| | \mathcal{H}_{(k-1)H}, \mathcal{E}_k^p]$$

$$\le \sum_{h=1}^H 2\Big( \sqrt{\frac{14S}{\max\{1, N_k(s_{kh}^p, \pi_{kp})\}} \log(\frac{2At_k}{\delta})} + \sqrt{\frac{7}{2 \max\{1, N(s_{kh}^p, \pi_{kp})\}} \log(\frac{2SAt_k}{\delta})} \Big)$$

where $N_k(s_{kh}^p, \pi_{kp}) = \sum_{i=1}^n N_k^i(s_{kh}^p, \pi_{kp})$ is the accumulated number of appearing times for the state-action pair $(s_{kh}^p, \pi_{kp})$ across all $n$ agents before period $k$.

Since $\bar{r} \in [0, 1]$, so that $\Delta_{kp} \le H$. Hence

$$\mathbb{E}[\Delta_{kp}^{\text{conc}}|\mathcal{H}_{(k-1)H}, \mathcal{E}_k^p] \leq \min\{H, \sum_{h=1}^{H} \beta_k(s_{kh}^p, \pi_{kp})\} \tag{D.5}$$

where

$$\beta_k(s_{kh}^p, \pi_{kp}) = 2\left(\sqrt{\frac{14S}{\max\{1, N_k(s_{kh}^p, \pi_{kp})\}} \log(\frac{2At_k}{\delta})} + \sqrt{\frac{7}{2\max\{1, N_k(s_{kh}^p, \pi_{kp})\}} \log(\frac{2SAt_k}{\delta})}\right) \tag{D.6}$$

Let $K = \lceil T/H \rceil$. Now in (D.6), we let

$$\delta = \frac{2}{Kn^2}$$

Then

$$\beta_k(s_{kh}^p, \pi_{kh}) \leq 2\sqrt{\frac{70S \log(2SAKt_k n)}{\max\{1, N_{t_k}(s_{kh}^p, \pi_{kp})\}}}$$

For this choice of $\delta$ and $\beta_k$, by summing over all state-action pairs, we have that

$$\mathbb{P}(M^* \notin M_k) < \sum_{s,a}\left(\frac{n\delta}{20SA} + \frac{n\delta}{60SA}\right) = \frac{n\delta}{15}$$

Using the fact that $\Delta_{kp} \leq H$ we can decompose regret as follows:

$$\sum_{p=1}^{n}\sum_{k=1}^{K}\Delta_{kp} \leq \sum_{p=1}^{n}\sum_{k=1}^{K}\Delta_{kp}\mathbf{1}(M^* \in \mathcal{M}_k, M_k^p \in \mathcal{M}_k) + H\sum_{p=1}^{n}\sum_{k=1}^{K}[\mathbf{1}(M_k^p \notin \mathcal{M}_k) + \mathbf{1}(M^* \notin \mathcal{M}_k)]$$

Using (D.3) and (D.5), we get that

$$\mathbb{E}[\sum_{p=1}^{n}\sum_{k=1}^{K}\Delta_{kp}] \leq \sum_{p=1}^{n}\sum_{k=1}^{K}\mathbb{E}[\Delta_{kp}\mathbf{1}(M_k^p \in \mathcal{M}_k, M^* \in \mathcal{M}_k)] + 2H\sum_{p=1}^{n}\sum_{k=1}^{K}\mathbb{P}(M^* \notin \mathcal{M}_k)$$

$$\leq H\mathbb{E}[\sum_{p=1}^{n}\sum_{k=1}^{K}\sum_{h=1}^{H}\min\{\beta_k(s_{t_k+h}^p, \pi_{kp}), 1\}] + 2HKn\frac{2n}{15Kn^2}$$

$$= H\mathbb{E}[\sum_{p=1}^{n}\sum_{k=1}^{K}\sum_{h=1}^{H}\min\{\beta_k(s_{t_k+h}^p, \pi_{kp}), 1\}] + \frac{4}{15}H$$

Also note that under the constraint that reward $\in [0,1]$, we have that

$$\sum_{p=1}^{n}\sum_{k=1}^{K}\Delta_{kp} \leq nT$$

So we are interested in providing a bound on

$$\min\{H\sum_{p=1}^{n}\sum_{k=1}^{K}\sum_{h=1}^{H}\min\{\beta_k(s_{kh}^p, \pi_{kp}), 1\} + \frac{4}{15}H, nT\} \tag{D.7}$$

Now consider the event $\mathcal{A}_k(s,a) = \{N_{t_k}(s,a) \leq Hn\}$. This event can happen less than or equal to $2Hn$ times per state-action pair of each agent throughout the whole process and across all agents. Suppose one state-action pair $(s,a)$ has appeared in the process for some agent, i.e. $(s,a) = (s^p_{kh}, \pi_{kp})$ for some $k, h, p$. Note that $t_k = (k-1)H + 1$ is the start time step of episode $k$ and $N_{t_k}(s,a)$ is the number of times that $(s,a)$ appears before $t_k$. Suppose $(s,a)$ has appeared for a number of times less than or equal to $Hn$, prior to the start time of episode $k$, and also note that $(s,a)$ appears at most $Hn$ more times within the new episode $k$. And once $(s,a)$ appears more than $(>) Hn$ times by the end of episode $k$, then event $\mathcal{A}_j(s,a)$ doesn't happen if $j \geq k+1$. And in this way the event will happen less than or equal to $2Hn$ times across the whole process and all agents. Therefore,

$$\sum_{p=1}^{n} \sum_{k=1}^{K} \sum_{h=1}^{H} \mathbf{1}(N^p_k(s^p_{kh}, \pi_{kp}) \leq Hn) \leq 2HnSA$$

Now we look at the complement event of $\mathcal{A}_k(s,a)$ where $\{N_{t_k}(s,a) > Hn\}$. Then for any $t \in \{t_k, \ldots, t_{k+1}-1\}$, $N_t(s,a) + n \leq N_{t_k}(s,a) + Hn \leq 2N_{t_k}(s,a)$. Therefore by a similar analysis from [16] we have

$$\sum_{p=1}^{n} \sum_{k=1}^{K} \sum_{t=t_k}^{t_{k+1}-1} \sqrt{\frac{\mathbf{1}(N_{t_k}(s^p_t, \pi_{kp}) > Hn)}{N_{t_k}(s^p_t, \pi_{kp})}} \leq \sum_{p=1}^{n} \sum_{k=1}^{K} \sum_{t=t_k}^{t_{k+1}-1} \sqrt{\frac{2}{N_t(s^p_t, \pi_{kp}) + n}} \tag{D.8}$$

$$= \sqrt{2} \sum_{p=1}^{n} \sum_{t=1}^{T} (N_t(s^p_t, \pi_{kp}) + n)^{-1/2} \tag{D.9}$$

$$\leq \sqrt{2} \sum_{s,a} \sum_{j=1}^{N_{T+1}(s,a)} j^{-1/2} \leq \sqrt{2} \sum_{s,a} \int_{x=0}^{N_{T+1}(s,a)} x^{-1/2} dx \tag{D.10}$$

$$\leq \sqrt{2SA \sum_{s,a} N_{T+1}(s,a)} = \sqrt{2SAnT} \tag{D.11}$$

Note that all the rewards and transitions are absolutely constrained in $[0,1]$. Using (D.5), and also using the fact that $n \leq O(S^2 AT \log(SAnT))$, (D.7) is bounded by:

$$\min\{H \sum_{p=1}^{n} \sum_{k=1}^{K} \sum_{h=1}^{H} \min\{\beta_k(s^p_{kh}, \pi_{kp}), 1\} + \frac{4}{15}H, nT\} \leq \min\{2H^2 nSA + 2H\sqrt{70S^2 AnT \log(SAnT)} + \frac{4}{15}H, Tn\}$$

$$\leq \sqrt{2H^2 SAnT} + 2H\sqrt{70S^2 AnT \log(SAnT)} + \frac{4}{15}H$$

$$\leq 12HS\sqrt{AnT \log(SAnT)} + \frac{4}{15}H$$

$$\leq \tilde{O}(HS\sqrt{AnT})$$

$\square$

# E  Proof of Finite-Horizon Case under Dirichlet Prior

## E.1  Proof of Theorem (3.2)

*Proof.* From Lemma (D.4), (D.5) and (D.6), under Dirichlet prior, by a union bound at each $x^p_{kh} = (s,a)$ for some state-action pair $(s,a)$ for $P(s,a)$ and $R(s,a)$, we have that for each $p$, with probability at least $1 - \frac{1}{nT}$,

$$\mathbb{E}[\sum_{h=1}^{H}\{|w^R(x_{kh}^p)| + |w_h^P(x_{k,h+1}^p)|\}|\mathcal{H}_{(k-1)H}] \le \sum_{h=1}^{H} 2(H+1)\sqrt{\frac{2\log(4SAnT)}{\max\{N_k(x_{kh}^p)-2,1\}}} \quad \text{(E.1)}$$

Let $\mathcal{A} = \{(s,a)|N(s,a) \le 2\}$ be the set of $(s,a)$ pairs which have only appear less than or equal to 2 times throughout the whole process. Note that for $(s,a) \in \mathcal{A}$, when these pairs appear, since the reward function is strictly in $[0,1]$, and there are at most $SA$ appears in total. So the total regret regarding these pairs is upper bounded by $2SA + 1$.

For the $x_{kh}^p \notin \mathcal{A}$ we may use (E.1) to bound as

$$\mathbb{E}[\sum_{h=1}^{H}\{|w^R(x_{kh}^p)| + |w_h^P(x_{k,h+1}^p)|\}|\mathcal{H}_{k1}] \le \sum_{h=1}^{H} 2(H+1)\sqrt{\frac{2\log(4SAnT)}{N_k(x_{kh}^p)-2}} \quad \text{(E.2)}$$

So we have

$$\text{BayesRegret}(T,\pi,\phi) \le \Big(\sum_{k=1}^{\lceil T/H \rceil}\sum_{p=1}^{n}\sum_{h=1}^{H} 2(H+1)\sqrt{\frac{2\log(4SAnT)}{N_k(x_{kh}^p)-2}} + 2SA+1\Big)\Big(1 - \frac{1}{Tn}\Big) + \frac{SATn}{Tn}$$

$$\le \sum_{k=1}^{\lceil T/H \rceil}\sum_{p=1}^{n}\sum_{h=1}^{H} 2(H+1)\sqrt{\frac{2\log(4SAnT)}{N_k(x_{kh}^p)-2}} + 3SA+1$$

We again define $\mathcal{A}_k(s,a) = \{N_k(s,a) \le Hn\}$. Then again for any $t \in \{t_k,\ldots,t_{k+1}-1\}$, $N_t(s,a) + n \le N_k(s,a) + Hn \le 2N_k(s,a)$. Then similar to (D.8), for $n > 4$

$$\sum_{p=1}^{n}\sum_{k=1}^{\lceil T/H \rceil}\sum_{h=1}^{H}\sqrt{\frac{\mathbf{1}(N_k(s_{kh}^p,\mu_{kp}) > Hn)}{N_k(s_{kh}^p,\mu_{kh}^p)-2}} \le \sum_{p=1}^{n}\sum_{k=1}^{\lceil T/H \rceil}\sum_{t=t_k}^{t_{k+1}-1}\sqrt{\frac{2}{N_t(s_t^p,\mu_{kp})+n-4}}$$

$$= \sqrt{2}\sum_{p=1}^{n}\sum_{t=1}^{T}(N_t(s_t^p,\mu_{kp})+n-4)^{-1/2}$$

$$\le \sqrt{2}\sum_{s,a}\sum_{j=1}^{N_{T+1}(s,a)-4} j^{-1/2} \le \sqrt{2}\sum_{s,a}\int_{x=0}^{N_{T+1}(s,a)-4} x^{-1/2}dx$$

$$\le \sqrt{2SA\sum_{s,a}(N_{T+1}(s,a)-4)} \le \sqrt{2SAnT}$$

$$\sum_{p=1}^{n}\sum_{k=1}^{m}\sum_{h=1}^{H}\mathbf{1}(N_k^p(s_{kh}^p,\mu_{kp}) \le Hn) \le 2HnSA$$

Also for $x_{kh}^p = (s_{kh}^p, a_{kh}^p) \in \mathcal{A}_k$, $|w^R(x_{kh}^p)| + |w_h^P(x_{k,h+1}^p)| \le (H+1)$. Hence we have

$$\text{BayesRegret}(T,\pi,\phi) \le \sum_{k=1}^{\lceil T/H \rceil}\sum_{p=1}^{n}\sum_{h=1}^{H} 2(H+1)\sqrt{\frac{2\log(4SAnT)}{N_k(x_{kh}^p)-2}} + 2SA+1$$

$$\le 2(H+1)\sqrt{4\log(4SAnT)SAnT} + 2H(H+1)SnA + 2SA + 1$$

$$\le \tilde{O}(H\sqrt{SAnT})$$

$\square$

# F  Proof of Infinite-Horizon Case under General Prior

In order to prove Theorem (4.1), we provide the following lemmas:

**Lemma F.1.** (Bound on number of epochs). When $Tn \geq SA$, the number of epochs $K$ of Algorithm 2 is upper bounded by

$$K \leq SA \log_2(\frac{8Tn}{SA})$$

**Lemma F.2.** Conditioning on prior distribution $\phi$,

$$\mathbb{E}[\sum_{k=1}^{K} \sum_{p=1}^{n} (Q_{s_t^p,a_t^p}^k - P_{s_t^p,a_t^p})^T \tilde{h}|\phi] \leq \tilde{O}(D\sqrt{SATn})$$

**Lemma F.3.** With probability at least $1 - 2\delta$,

$$\sum_{t=\tau_k}^{\tau_{k+1}} \sum_{p=1}^{n} (P_{s_t^p,a_t^p} - \mathbf{1}_{s_t^p})^T \tilde{h} \leq O\left(D\sqrt{n(\tau_{k+1} - \tau_k) \log(n/\delta)}\right)$$

## F.1  Proof of Lemma (F.1)

*Proof.* We use the technique similar to [4] to give the upper bound for the number of epochs. Define $N(s,a) = \{t < T+1 : s_t = s, a_t = a\}$ be the total number of observations of the state-action pair $(s,a)$ up to time step $T$. Let $v_k(s,a)$ be the number of observations of state-action pair $(s,a)$ within epoch $k$. For each epoch $k < K$ there exists a state-action pair $(s,a)$ with $v_k(s,a) = N_{s,a}^{\tau_k}$, or $v_k(s,a) = 1, N_{s,a}^{\tau_k} = 0$. Let $K(s,a)$ be the number of epochs with $v_k(s,a) = N_{s,a}^{\tau_k}$ and $N_{s,a}^{\tau_k} > 0$. If $N(s,a) > 0$, then $v_k(s,a) = N_{s,a}^{\tau_k}$ implies $N_{s,a}^{\tau_{k+1}} = 2N_{s,a}^{\tau_k}$. so that

$$N(s,a) = \sum_{k=1}^{K} v_k(s,a) \geq 1 + \sum_{k:v_k(s,a)=N_{s,a}^{\tau_k}} N_{s,a}^{\tau_k} \geq 1 + \sum_{i=1}^{K(s,a)} 2^{i-1} = 2^{K(s,a)}$$

If $N(s,a) = 0$, then $K(s,a) = 0$, hence $N(s,a) \geq 2^{K(s,a)} - 1$ for any state-action pair $(s,a)$. It follows that

$$Tn = \sum_{s,a} N(s,a) \geq \sum_{s,a} (2^{K(s,a)} - 1) \tag{F.1}$$

In each epoch $k$, note that there exists a state-action pair $(s,a)$, such that it's visited with either $N_{s,a}^{\tau_k} = 0$ or $N_{s,a}^{\tau_k} = v_k(s,a)$. So we have

$$K \leq 1 + SA + \sum_{s,a} K(s,a) \Rightarrow \sum_{s,a} K(s,a) \geq K - 1 - SA$$

This implies

$$\sum_{s,a} 2^{K(s,a)} \geq SA 2^{\sum_{s,a} K(s,a)/SA} \geq SA 2^{\frac{K-1}{SA} - 1}$$

With (F.1) we have:

$$Tn \geq SA(2^{\frac{K-1}{SA} - 1} - 1)$$

So we have

$$K \leq 1 + 2SA + SA \log_2(\frac{Tn}{SA})$$

So when $Tn \geq SA$, we have

$$K \leq SA \log_2(\frac{8Tn}{SA})$$

$\square$

## F.2 Proof of Lemma (F.2)

*Proof.* Similar to the proof of Theorem (3.1), define $\hat{P}_a^t(\cdot|s)$ as the empirical distribution up to time $t$ steps after sampling $(s, a)$, define $\hat{R}_a^t(s)$ as the empirical average reward. Define $N_k(s, a) = \sum_{t=1}^{\tau_k - 1} \mathbf{1}[(s_t, a_t) = (s, a)]$ to be the number of times $(s, a)$ was sampled prior to time $\tau_k$, where $\tau_k$ is the start time step of epoch $k$. Empirical estimate of rewards and transition probabilities at each $(s, a)$ are defined as

$$\hat{R}_a^{\tau_k}(s) = \frac{R_k(s, a)}{\max\{1, N_k(s, a)\}} \quad \hat{P}_a^{\tau_k}(s'|s) = \frac{P_k(s, a, s')}{\max\{1, N_k(s, a)\}}$$

where

$$R_k(s, a) = \sum_{\tau=1}^{\tau_k - 1} r_\tau \mathbf{1}[s_\tau = s, a_\tau = a]; \quad P_k(s, a, s') = \#\{\tau < \tau_k : s_\tau = s, a_\tau = a, s_{\tau+1} = s'\}$$

Define the confidence set for episode $k$:

$$\mathcal{M}_k = \{M : \left\|\hat{P}_a^{\tau_k}(\cdot|s) - P_a^M(\cdot|s)\right\|_1 \leq B_{k1}, \ |\hat{R}_a^{\tau_k}(s) - R_a^M(s)| \leq B_{k2} \ \forall(s, a)\} \tag{F.2}$$

where

$$B_{k1} = \sqrt{\frac{14S}{\max\{1, N_k(s_{kh}^p, \mu_{kp})\}} \log(\frac{2A\tau_k}{\delta})}$$

$$B_{k2} = \sqrt{\frac{7}{2\max\{1, N(s_{kh}^p, \mu_{kp})\}} \log(\frac{2SA\tau_k}{\delta})}$$

One key step for analyzing the Bayesian regret bound of posterior sampling is to notice that conditioning on $\mathcal{F}_{k-1}$, we have

$$M^*|\mathcal{F}_{k-1} \stackrel{D}{=} M_k^p|\mathcal{F}_{k-1} \ \forall p$$

where $M^*$ is the true MDP, and $M_k^p$ is the sampled MDP by agent $p$. Similar to the proof of Theorem (3.1), at time step $t$, $N_t(s, a)$ takes possible values over $1, 2, \ldots, n(t-1)$, where $n$ is the number of agents. Define

$$\mathcal{E}_k^p = \{M^* \in \mathcal{M}_k, M_k^p \in \mathcal{M}_k\}$$

So by a similar union bound idea in the proof of Theorem (3.1) and equation (F.2), we have that

$$\mathbb{E}[(Q_{s_t^p, a_t^p}^k - \hat{P}_{s_t^p, a_t^p}^{\tau_k})^T \tilde{h}|\mathcal{F}_{k-1}, \mathcal{E}_k^p] \leq 2D\sqrt{\frac{14S}{\max\{1, N_k(s_t^p, a_t^p)\}} \log(\frac{2SA\tau_k}{\delta})} \tag{F.3}$$

$$\mathbb{E}[(\hat{P}_{s_t^p, a_t^p}^{\tau_k} - P_{s_t^p, a_t^p})^T \tilde{h}|\mathcal{F}_{k-1}, \mathcal{E}_k^p] \leq 2D\sqrt{\frac{14S}{\max\{1, N_k(s_t^p, a_t^p)\}} \log(\frac{2SA\tau_k}{\delta})} \tag{F.4}$$

And

$$\mathbb{P}(M^* \notin \mathcal{M}_k) \leq \frac{n\delta}{15}$$

Also note that for any $t \in \{\tau_k, \ldots, \tau_{k+1} - 1\}$, $N_t(s, a) \leq 2N_k(s, a)$, so we have that for fixed $\delta > 0$ in the range where the probabilities above are well defined,

$$\sum_{k=1}^{K} \sum_{p=1}^{n} \mathbb{E}[(Q_{s_t^p, a_t^p}^k - P_{s_t^p, a_t^p})^T \tilde{h} | \mathcal{F}_{(k-1)H}] = \mathbb{E}[\sum_{k=1}^{K} \sum_{p=1}^{n} (Q_{s_t^p, a_t^p}^k - \hat{P}_{s_t^p, a_t^p}^{\tau_k})^T \tilde{h} | \mathcal{F}_{k-1}]$$

$$+ \mathbb{E}[\sum_{k=1}^{K} \sum_{p=1}^{n} (\hat{P}_{s_t^p, a_t^p}^{\tau_k} - P_{s_t^p, a_t^p})^T \tilde{h} | \mathcal{F}_{k-1}]$$

$$\leq \sum_{k=1}^{K} \sum_{p=1}^{n} \left( 4D \sqrt{\frac{14S}{\max\{1, N_k(s_t^p, a_t^p)\}} \log(\frac{2SA\tau_k}{\delta})} + \frac{4Dn\delta}{15} \right)$$

$$\leq 4D \sqrt{14S \log(\frac{2SAT}{\delta})} \sum_{k=1}^{K} \sum_{p=1}^{n} \sum_{t=\tau_k}^{\tau_{k+1}-1} \sqrt{\frac{2}{\max\{1, N_t(s_t^p, a_t^p)\}}} + \frac{4DKn^2\delta}{15}$$

$$= 4D \sqrt{14S \log(\frac{2SAT}{\delta})} \sum_{p=1}^{n} \sum_{t=1}^{T} \sqrt{\frac{2}{\max\{1, N_t(s_t^p, a_t^p)\}}} + \frac{4DKn^2\delta}{15}$$

$$\leq 4D \sqrt{14S \log(\frac{2SAT}{\delta})} \sqrt{2} \sum_{s,a} \sum_{j=1}^{N_{T+1}(s,a)} j^{-1/2} + \frac{4DKn^2\delta}{15}$$

$$\leq 4D \sqrt{14S \log(\frac{2SAT}{\delta})} \sqrt{2SAnT} + \frac{4DKn^2\delta}{15}$$

Taking $\delta = \frac{1}{n^2 K}$, then we have

$$\sum_{k=1}^{K} \sum_{p=1}^{n} \mathbb{E}[(Q_{s_t^p, a_t^p}^k - P_{s_t^p, a_t^p})^T \tilde{h} | \mathcal{F}_{k-1}] \leq \tilde{O}(DS \sqrt{ATn})$$

So we have the regret bound when conditioning on prior distribution $\phi$.

$\square$

### F.3 Proof of Lemma (F.3)

*Proof.* We expand the left hand side of (G.2) as

$$\sum_{t=\tau_k}^{\tau_{k+1}-1} \sum_{p=1}^{n} (P_{s_t^p, a_t^p} - \mathbf{1}_{s_t^p})^T \tilde{h} = \sum_{t=\tau_k}^{\tau_{k+1}-1} \sum_{p=1}^{n} [P_{s_t^p, a_t^p} - \mathbf{1}_{s_{t+1}^p} + \mathbf{1}_{s_{t+1}^p} - \mathbf{1}_{s_t^p}]^T \tilde{h}$$

$$= \underbrace{\sum_{t=\tau_k}^{\tau_{k+1}-1} \sum_{p=1}^{n} (P_{s_t^p, a_t^p} - \mathbf{1}_{s_{t+1}^p})^T \tilde{h}}_{(1)} + \underbrace{\sum_{t=\tau_k}^{\tau_{k+1}-1} \sum_{p=1}^{n} (\mathbf{1}_{s_{t+1}^p} - \mathbf{1}_{s_t^p})^T \tilde{h}}_{(2)}$$

**Bound on part (1):**

Let $Z_p = \sum_{t=\tau_k}^{\tau_{k+1}-1} (P_{s_t^p, a_t^p} - \mathbf{1}_{s_{t+1}^p})^T \tilde{h}$ for each $p$. Let $\mathcal{E}_k$ be the event defined as

$$\mathcal{E}_{\delta}^k = \{ |Z_p| \leq D \sqrt{8(\tau_{k+1} - \tau_k) \log(4n/\delta)}, \forall p \}$$

Let

$$A_{\delta}^k = \{ \sum_{t=\tau_k}^{\tau_{k+1}-1} \sum_{p=1}^{n} (P_{s_t^p, a_t^p} - \mathbf{1}_{s_{t+1}^p})^T \tilde{h} \geq 4D \sqrt{n(\tau_{k+1} - \tau_k)} \log(\frac{\sqrt{8n}}{\delta}) \}$$

Thus

$$\mathbb{P}[\sum_{t=\tau_k}^{\tau_{k+1}-1}\sum_{p=1}^{n}(P_{s_t^p,a_t^p}-\mathbf{1}_{s_{t+1}^p})^T\tilde{h} \geq 4D\sqrt{n(\tau_{k+1}-\tau_k)}\log(\frac{\sqrt{8n}}{\delta})] = \mathbb{P}(A_\delta^k,\mathcal{E}_\delta^k) + \mathbb{P}(A_\delta^k,(\mathcal{E}_\delta^k)^c)$$

$$\leq \mathbb{P}(A_\delta^k,\mathcal{E}_\delta^k) + \mathbb{P}((\mathcal{E}_\delta^k)^c)$$

Note that for any $p,t$,

$$\mathbb{E}[\mathbf{1}_{s_{t+1}^p}^T\tilde{h}|\tilde{\pi}_k,\tilde{h},s_t^p] = P_{s_t^p,a_t^p}^T\tilde{h} \tag{F.5}$$

On event $\mathcal{E}_\delta^k$ we have that

$$\mathbb{P}(A_\delta^k,\mathcal{E}_\delta^k) = \mathbb{P}(A_\delta^k|\mathcal{E}_\delta^k)\mathbb{P}(\mathcal{E}_\delta^k) \leq \mathbb{P}(A_\delta^k|\mathcal{E}_\delta^k)$$

Note that by the fact that $\sqrt{ab} \leq \frac{a+b}{2}$ for $a,b > 0$, we have

$$\sqrt{\log(\frac{2}{\delta})\log(\frac{4n}{\delta})} \leq \frac{1}{2}(\log(\frac{2}{\delta})+\log(\frac{4n}{\delta})) = \log(\frac{\sqrt{8n}}{\delta})$$

By (F.5), $(P_{s_t^p,a_t^p}-\mathbf{1}_{s_{t+1}^p})^T\tilde{h}$ is a martingale difference sequence for each $p$. So by Hoeffding-Azuma inequality,

$$\mathbb{P}(A_\delta^k|\mathcal{E}_\delta^k) = \mathbb{P}(\sum_{t=\tau_k}^{\tau_{k+1}-1}\sum_{p=1}^{n}(P_{s_t^p,a_t^p}-\mathbf{1}_{s_{t+1}^p})^T\tilde{h} \geq 4D\sqrt{n(\tau_{k+1}-\tau_k)}\log(\frac{\sqrt{8n}}{\delta}))$$

$$\leq \mathbb{P}(\sum_{t=\tau_k}^{\tau_{k+1}-1}\sum_{p=1}^{n}(P_{s_t^p,a_t^p}-\mathbf{1}_{s_{t+1}^p})^T\tilde{h} \geq 4D\sqrt{n(\tau_{k+1}-\tau_k)}\sqrt{\log(\frac{2}{\delta})\log(\frac{4n}{\delta})})$$

$$\leq \frac{\delta}{2}$$

Again by Hoeffding-Azuma inequality and the fact that the diameter of the sampled MDP is bounded by $2D$, $Z_p$ are i.i.d. across $p$, we have

$$\mathbb{P}((\mathcal{E}_\delta^k)^c) \leq \sum_{p=1}^{n}\mathbb{P}(|Z_p| \geq D\sqrt{8(\tau_{k+1}-\tau_k)\log(4n/\delta)}) \leq 2n\frac{\delta}{4n} = \frac{\delta}{2}$$

Hence

$$\mathbb{P}(A_\delta^k) \leq \mathbb{P}(A_\delta^k|\mathcal{E}_\delta^k) + \mathbb{P}((\mathcal{E}_\delta^k)^c) \leq \frac{\delta}{2} + \frac{\delta}{2} = \delta$$

**Bound on part (2):**

$$(2) = \sum_{t=\tau_k}^{\tau_{k+1}-1}\sum_{p=1}^{n}(\mathbf{1}_{s_{t+1}^p}-\mathbf{1}_{s_t^p})^T\tilde{h}$$

$$= \sum_{p=1}^{n}(\mathbf{1}_{s_{\tau_{k+1}}^p}-\mathbf{1}_{s_{\tau_k}^p})^T\tilde{h}$$

Note that $(\mathbf{1}_{s^p_{\tau_{k+1}}} - \mathbf{1}_{s^p_{\tau_k}})^T \tilde{h}$ are. i.i.d. across $p = 1, 2, \ldots, n$, and that

$$|(\mathbf{1}_{s^p_{\tau_{k+1}}} - \mathbf{1}_{s^p_{\tau_k}})^T \tilde{h}| \leq 2D$$

Hence using Azuma-Hoeffding inequality, we have that for all $\epsilon > 0$,

$$\mathbb{P}(\sum_{p=1}^n (\mathbf{1}_{s^p_{\tau_{k+1}}} - \mathbf{1}_{s^p_{\tau_k}})^T \tilde{h} \geq \epsilon) \leq \exp(-\frac{\epsilon^2}{8nD^2})$$

Taking $\epsilon = D\sqrt{8n \log(1/\delta)}$ in the above, then we have that with probability at least $1 - \delta$,

$$\sum_{p=1}^n (\mathbf{1}_{s^p_{\tau_{k+1}}} - \mathbf{1}_{s^p_{\tau_k}})^T \tilde{h} \leq D\sqrt{8n \log(1/\delta)} \tag{F.6}$$

$\square$

### F.4 Proof of Theorem (4.1)

*Proof.* Note that

$$\sum_k \tau_{k+1} - \tau_k \leq T \quad \text{and} \quad \sum_{k=1}^K \sqrt{\tau_{k+1} - \tau_k} \leq \sqrt{KT}$$

From the above Lemmas in Appendix C, and by taking $\delta = 1/(Tn)$ in Lemma (F.2, F.3), we have that conditional on prior $\phi$

$$\text{BayesRegret}(T, \mathcal{M}, n, \phi) \leq \tilde{O}(DS\sqrt{ATn})$$

$\square$

## G Proof of Infinite-Horizon Case under Dirichlet Prior

### G.1 Useful Lemmas

**Lemma G.1. (Azuma-Hoeffiding inequality [12])** Let $X_1, X_2, \ldots$ be a martingale difference sequence with $|X_i| \leq c$ for all $i$. Then for all $\epsilon > 0$ and $n \in \mathbb{N}$

$$\mathbb{P}(\sum_{i=1}^n X_i \geq \epsilon) \leq \exp(-\frac{\epsilon^2}{2nc^2})$$

**Lemma G.2. (Transition concentration).** Let $\tilde{P}_{s,a} = Q^k_{s,a}$ in Algorithm 2. Let $\hat{P}_{s,a} = \mathbb{E}[P_{s,a}|\mathcal{H}_{k1}]$, then for any independent prior over rewards with $r \in [0, 1]$, additive sub-Gaussian noise and an independent Dirichlet prior over transitions, then for any state-action pair $(s, a)$ in episode $k$, with probability $1 - \delta$,

$$|(\tilde{P}_{s,a} - \hat{P}_{s,a})^T \tilde{h}| \leq O(D\sqrt{n \log(2/\delta)/N_k(s, a)})$$

### G.2 Proof of Theorem (4.2)

*Proof.* As defined before,

$$\mathcal{R}(T, \mathcal{M}, n, \phi) = \mathbb{E}[Tn\lambda^* - \sum_{p=1}^n \sum_{t=1}^T r(s^p_t, a^p_t)|\mathcal{M}^* \sim \phi]$$

$\lambda^*$ is the optimal gain of MDP $\mathcal{M}$. Algorithm (2) proceeds in epochs $k = 1, 2, \ldots, K$, where $K \leq SA \log(T)$. Now we define

$$R_k = (\tau_{k+1} - \tau_k)n\lambda^* - \sum_{t=\tau_k}^{\tau_{k+1}-1} \sum_{p=1}^{n} r_{s_t^p, a_t^p}$$

So

$$\mathcal{R}(T, \mathcal{M}, n, \phi) = \mathbb{E}[\sum_{k=1}^{K} R_k | \mathcal{M}^* \sim \phi]$$

We can write $R_k$ as

$$R_k = \sum_{t=\tau_k}^{\tau_{k+1}-1} \sum_{p=1}^{n} [\lambda^* - r_{s_t^p, a_t^p}]$$

$$= \sum_{t=\tau_k}^{\tau_{k+1}-1} \sum_{p=1}^{n} [(\lambda^* - \tilde{\lambda}_k) + (\tilde{\lambda}_k - r_{s_t^p, a_t^p})]$$

where $\tilde{\lambda}_k$ is the optimal gain of the sampled MDP $\tilde{M}^k$. From Lemma (2.1) we know that for any state $s$, optimal policy $\tilde{\pi}_k$ for communicating MDP $\tilde{M}^k$, action $a = \tilde{\pi}_k(s)$, $\lambda_k = r_{s,a} + \tilde{P}_{s,a}^T \tilde{h} - \tilde{h}_s$, where $\tilde{P}_{s,a} = Q_{s,a}^k$.

One key property of Thompson sampling is that, conditional upon the data $\mathcal{F}_{k-1}$, the transitions are independent of the transitions sampled by Thompson sampling. Let

$$\hat{P}_{s,a} = \mathbb{E}[P_{s,a} | \mathcal{F}_{k-1}]$$

By Lemma G.2, for any fixed $(s, a)$ in episode $k$, with probability at least $1 - \delta$,

$$|(\tilde{P}_{s,a} - \hat{P}_{s,a})^T \tilde{h}| \leq O\Big(2D\sqrt{2\log(2/\delta)/N_k(s,a)}\Big)$$

Which indicates

$$|\mathbb{E}[(\tilde{P}_{s,a} - P_{s,a})^T \tilde{h} | \mathcal{F}_{k-1}]| \leq O\Big(2D\sqrt{2\log(2/\delta)/N_k(s,a)}\Big)$$

By the proof of Lemma 5.4 in [3], we can have that with probability $1 - \delta$, for every epoch $k$, the optimal gain $\tilde{\lambda}_k$ of the MDP $\tilde{M}_k$ satisfies:

$$\mathbb{E}[(\lambda^* - \tilde{\lambda}_k) | \mathcal{F}_{k-1}] \leq O\Big(2D\sqrt{2\log(2/\delta)/N_k(s,a)}\Big)$$

where $N_k(s, a)$ is the sample size of some pair of state-action pair $(s, a)$. Hence by Cauchy-Schwarz's inequality, with probability $1 - \delta$,

$$\sum_{k=1}^{K} \mathbb{E}[\sum_{t=\tau_k}^{\tau_{k+1}-1} \sum_{p=1}^{n} (\lambda^* - \tilde{\lambda}_k) | \mathcal{F}_{k-1}] \leq \sum_{k=1}^{K} \sum_{t=\tau_k}^{\tau_{k+1}-1} \sum_{p=1}^{n} 2D\sqrt{\frac{2\log(2/\delta)}{N_k(s,a)}}$$

$$\leq 2D\sqrt{2\log(2/\delta)}\sqrt{\sum_{s,a} N_k(s,a)}$$

$$\leq 2D\sqrt{2\log(2/\delta)}\sqrt{SATn}$$

Hence

$$\sum_{k=1}^{K} \mathbb{E}[\sum_{t=\tau_k}^{\tau_{k+1}-1} \sum_{p=1}^{n} (\lambda^* - \tilde{\lambda}_k) | \mathcal{F}_{k-1}] \leq O(D\sqrt{nTSA}) \tag{G.1}$$

$$\sum_{t=\tau_k}^{\tau_{k+1}-1} \sum_{p=1}^{n} (\tilde{\lambda}_k - r_{s_t^p, a_t^p}) = \sum_{t=\tau_k}^{\tau_{k+1}-1} \sum_{p=1}^{n} (\tilde{P}_{s_t^p, a_t^p} - \mathbf{1}_{s_t^p})^T \tilde{h}$$

$$= \sum_{t=\tau_k}^{\tau_{k+1}-1} \sum_{p=1}^{n} (\tilde{P}_{s_t^p, a_t^p} - P_{s_t^p, a_t^p} + P_{s_t^p, a_t^p} - \mathbf{1}_{s_t^p})^T \tilde{h}$$

Again, we use Azuma-Hoeffding inequality to obtain that with probability $1 - 2\rho$,

$$\sum_{t=\tau_k}^{\tau_{k+1}} \sum_{p=1}^{n} (P_{s_t^p, a_t^p} - \mathbf{1}_{s_t^p})^T \tilde{h} \leq O\left(D\sqrt{n(\tau_{k+1} - \tau_k)\log(n/\rho)}\right) \tag{G.2}$$

Similar to previous proof, by noting that

$$\sum_k \tau_{k+1} - \tau_k \leq T \quad \text{and} \quad \sum_{k=1}^{K} \sqrt{\tau_{k+1} - \tau_k} \leq \sqrt{KT}$$

And similar to Lemma F.1,

$$K \leq SA \log_2\left(\frac{8Tn}{SA}\right)$$

So

$$\sum_{k=1}^{K} \sum_{t=\tau_k}^{\tau_{k+1}} \sum_{p=1}^{n} (P_{s_t^p, a_t^p} - \mathbf{1}_{s_t^p})^T \tilde{h} \leq O(D\sqrt{nTSA\log(nT)\log(n)})$$

Hence

$$\sum_{k=1}^{K} \sum_{t=\tau_k}^{\tau_{k+1}} \sum_{p=1}^{n} (P_{s_t^p, a_t^p} - \mathbf{1}_{s_t^p})^T \tilde{h} \leq \tilde{O}(D\sqrt{nTSA})$$

As for $\sum_{t=\tau_k}^{\tau_{k+1}} \sum_{p=1}^{n} (\tilde{P}_{s_t^p, a_t^p} - P_{s_t^p, a_t^p})^T \tilde{h}$, by Lemma G.2, we have

$$(\tilde{P}_{s_t^p, a_t^p} - \hat{P}_{s_t^p, a_t^p})^T \tilde{h} \leq 2D\sqrt{2\log(2/\delta)/N_k(s_t^p, a_t^p)}$$

Again by Cauchy-Schwarz

$$\sum_{k=1}^{K} \sum_{t=\tau_k}^{\tau_{k+1}} \sum_{p=1}^{n} (\tilde{P}_{s_t^p, a_t^p} - \hat{P}_{s_t^p, a_t^p})^T \tilde{h} \leq O(D\sqrt{nTSA})$$

Which indicates

$$\sum_{k=1}^{K} \mathbb{E}[\sum_{t=\tau_k}^{\tau_{k+1}} \sum_{p=1}^{n} (\tilde{P}_{s_t^p, a_t^p} - P_{s_t^p, a_t^p})^T \tilde{h} | \mathcal{F}_{k-1}] \leq O(D\sqrt{nTSA})$$

Thus

$$\sum_{k=1}^{K} \mathbb{E}[\sum_{t=\tau_k}^{\tau_{k+1}} \sum_{p=1}^{n} (\tilde{\lambda}_k - r_{s_t^p, a_t^p}) | \mathcal{F}_{k-1}] \leq \tilde{O}(D\sqrt{nTSA}) \tag{G.3}$$

(G.1) and (G.3) indicate that

$$\text{BayesRegret}(T, \mathcal{M}, n, \phi) = \tilde{O}(D\sqrt{SATn})$$

$\square$

### G.3 Proof of Lemma (G.2)

*Proof.*

$$|(\tilde{P}_{s,a} - \hat{P}_{s,a})^T \tilde{h}| \leq D|(\tilde{P}_{s,a} - \hat{P}_{s,a})|$$

By Gaussian-Dirichlet dominance theorem (Lemma 2 in [21]), for any $\alpha \in \mathbb{R}_+$, with $\alpha^T \mathbf{1} \geq 2$, the random variables $\tilde{P}_{s,a} \sim \text{Dirichlet}(\alpha)$ and $X \sim \mathcal{N}(0, \sigma^2 = 1/\alpha^T \mathbf{1})$ are ordered,

$$X \succeq_{so} \tilde{P}_{s,a} - \hat{P}_{s,a} \Rightarrow |X|D \succeq_{so} |\tilde{P}_{s,a} - \hat{P}_{s,a}|D$$

So this result follows from Lemma 1 in [21], also Lemma (D.4) □