# OpenReview forum: "Society of Agents: Regret Bounds of Concurrent Thompson Sampling"
_NeurIPS.cc/2022/Conference — NeurIPS 2022 Accept_

### Official Review · Reviewer_WDrE · 2022-07-07

**Rating:** 6
**Confidence:** 4
**Soundness:** 3 good
**Presentation:** 3 good
**Contribution:** 2 fair

**Summary:**

This paper studies efficient Thompson sampling algorithm for concurrent RL. They mainly consider efficient learning in episodic setting and infinite-horizon setting. For both settings, they provide efficient learning algorithm for general prior and Dirichlet prior. Their regret bounds show that $\sqrt{n}$ speed up is possible when there are $n$ agents sharing the information and exploring the environment simultanously.

**Questions:**

It would be helpful for evaluation if the authors could clearly explain the technical difficulties of concurrent PSRL and the methods they used to tackle them.

**Limitations:**

The authors have explained several limitations in the conclusion sections.Other possible directions for improvements are listed as follows:
- This paper does not tackle the communication cost in concurrent learning. It would be interesting to design efficient learning algorithms with efficient communication (e.g. [1,2]).
- There are also works studying efficient PSRL with function approximation. I wonder whether $\sqrt{n}$ speed-up is also possible in the function approximation setting.

[1] Agarwal et al. Communication efficient parallel reinforcement learning.

[2] Wang et al. Distributed Bandit Learning: Near-Optimal Regret with Efficient Communication.

**Strengths And Weaknesses:**

Strengths:

To the best of my knowledge, this is the first work studying efficient Thompson sampling algorithm in concurrent setting. The algorithm and analysis is a natural extension of the previous work in the single-agent setting. The paper is well-written and the theoretical results are solid and reasonable.

Weakness:

The authors fail to explain the technical difficulties of concurrent Thompson sampling compared with the single-agent setting. It seems that the algorithm and analysis almost directly follows that in [1,2]. Since the main contribution of this work is their theoretical improvement, I wonder whether there is anything novel in the algorithmic design and the proof technique.



[1] Osband et al. Why is posterior sampling better than optimism for reinforcement learning?

[2] Agrawal et al. Posterior sampling for reinforcement learning: worst-case regret bounds.

---

> ### Author Response · Authors · 2022-08-02
> **Thank you for your review and positive evaluation!**
>
> Regarding the technical difficulty, we certainly should have provided more discussions to make it clear.  While some parts of the proofs share the similar analyses in [1,2] (as the single-agent base algorithm is the same), a more refined characterization is needed on the part that ensures the cooperative exploration does not lead to a linear blowup in total regret. Note that a direct extension of [1,2]'s arguments would result in a trivial O(n) bound. To do so, for the finite-horizon setting, we need to bound the trade-off between more data (induced by the sharing among agents) and more sub-optimality induced by more agents performing sub-optimal policy (during exploration). We start from the first principles and provide a direct analysis by using concentration inequalities on martingale difference sequences. Furthermore, the infinite-horizon setting is even more challenging because a priori it is also unclear at what frequency the data-sharing should be performed in order to retain the optimal O(\sqrt{n}) scaling. This is not an issue for the finite-horizon setting because the natural breakdown of episodes suggests that all agents can share data and recompute their policies at the end of each episode. However, in infinite horizon setting, each agent only has a single trajectory that never ends, so algorithmically, it is already not obvious. We show that doubling the sharing horizon each time would yield the O(\sqrt{n}) scaling (in fact, in our first attempt, we investigated the even divide similar to the episodic case, which yields linear blowup. Note that the difference is that the episodic case, the environment resets at the end of an episode, which does not happen in the infinite-horizon setting). We will update the camera-ready version to include this discussion, which certainly helps on clarifying the theoretical and algorithmic difficulty of our contributions. Thank you for the suggestion!
>
> Regarding communications cost and function approximation, both are excellent points, which we will investigate for the future (our conjecture for the function approximation is that the O(\sqrt{n}) scaling still holds, but the setting will be much beyond the current scope). Thank you also for bringing the two references to our attention, which we will acknowledge in our camera-ready version!

---

### Official Review · Reviewer_qFko · 2022-07-08

**Rating:** 6
**Confidence:** 3
**Soundness:** 3 good
**Presentation:** 3 good
**Contribution:** 2 fair

**Summary:**

The paper considers a concurrent RL setup, where a set of agents share experience when individually interacting in the same environment, and optimizing the same reward function. Thompson Sampling (TS) based exploration is considered and corresponding regret bounds are derived for the finite and infinite horizon setting. These bounds extend the TS guarantees for the single-agent RL case to the concurrent RL setup. Moreover, simple simulations are performed showing the computed regrets are of the same order of the obtained bounds.

**Questions:**

 Even if agents have different reward functions, I believe the power of cooperation should still be visible because they interact with the same environment. In such a case:
- I expect some of the guarantees to go through, but how would the overall results change?
- UCRL (or similar) approaches should ensure increased exploration w.r.t. when agents have the same reward. Do you think there is a right measure (in the rewards' space) to quantify this?

**Limitations:**

I don't foresee potential negative societal impacts.

**Strengths And Weaknesses:**

I have appreciated the motivations for the concurrent RL setup provided in the introduction, and I think the considered problem is interesting and relevant. Moreover, the paper is nicely written and the derived theoretical results are sound both intuitively and technically.
However, I do have a concern:

- I think most (if not all) of the claims made are under the assumption that the agents reward functions are the same?  This should be clarified better in the introduction.

---

> ### Author Response · Authors · 2022-08-02
> **Thank you for your review and positive evaluation!**
>
> First, thank you for bringing the point of the reward function up! Your understanding is correct and we will emphasize it better in the introduction: right now this is only discussed in the formulation section, both in the finite-horizon and infinite-horizon subsections; and we certainly agree with you in that discussing it early in the introduction again helps make it more clear).
>
> Regarding different reward functions, I think you have really good intuition: since the underlying transition probability stays the same, this could in principle be harnessed to crystallize a certain degree of cooperative effects. In terms of specific bounds, prompted by your insight, we have thought about this question a bit more, and our results can generalize without change to the setting where each agent's reward function r(s,a) is known (i.e. only the transitions are unknown). This setting is already applicable in certain applications where the dynamics are the main unknown. Additionally, if r(s,a) is unknown and different for different agents, then our conjecture is that the regret bound would generalize to this setting but with an additional factor of O(nSA) (but we haven't nailed down a detailed proof yet). We also think it might be interesting to look at structured rewards: where each agent's reward has two components, one shared by all agents, and the other that is different and unique to this agent. Finally, regarding UCRL, we again think you have a good insight and we share that same intuition. In the limited time, we do not yet have a concrete, quantitative answer. We leave the investigations for future work. Thank you for all the thought-provoking comments and we will add them in the conclusion for the camera-ready!

---

### Official Review · Reviewer_vsWP · 2022-07-09

**Rating:** 6
**Confidence:** 4
**Soundness:** 3 good
**Presentation:** 3 good
**Contribution:** 3 good

**Summary:**

This paper studies the Bayesian regret bound of Concurrent Thompsom Sampling, where n agents interact with the environment simultaneously. The existing works are confined to the optimism-in-face-of-uncertainty principle (UCB). This paper presents the first attempt to analyze posterior sampling based algorithms, both in the episodic and infinite-horizon setting.



**Questions:**

See weakness part

**Limitations:**

yes

**Strengths And Weaknesses:**

Strengths:
1 This paper provides the first line of theoretical analysis for TS-based algorithm in the concurrent episodic/infinite-horizon RL setting.
2 While the analysis techniques seem to follow from the seminal works of Benjamin Van Roy, the resulting bounds are good as they are minimax-optimal.
3 The theoretical guarantee is verified by simulation results, which is appreciated as empirical study has been largely ignored in the RL theory community.

Weakness:
1 Comparison to existing works. The paper motivates the use of TS by some empirical works, showing that TS can have advantage as agents can determine their policies individually, while UCB-type algorithms tend to use the same policy. (Line 72-77) This is a good motivation for the use of TS. However, it seems that the analysis of this paper does not verify this potential advantage over UCB-type algorithms theoretically. In particular, the paper does not contain a comparison between them. Are they all minimax-optimal? What is the advantages of TS-based algorithms over a UCB-based one?

2 Simulation result. While it is good to see that the paper contains simulation result, the experiments are not satisfactory as they do not include the UCB-type algorithms as baseline. While I understand that a thoroughly comprehensive comparison between UCB and TS in theroy may need more involved analysis, I strongly recommend the authors to conduct simulations to verify the effectiveness of TS.

I am willing to raise my score if my major concerns could be addressed.
---------------------- update after rebuttal
I have read the responses from the authors and increase the score to support the acceptance of this work.

---

> ### Author Response · Authors · 2022-08-02
> **Thank you for your review and constructive evaluation!**
>
> First, regarding the theoretical comparison between UCB and TS, the theoretical regret bounds alone will not be a good indicator to distinguish between the two. For instance, in single-agent settings, UCB and TS achieve the same minimax optimal regret bound in many different RL problems (including bandits, contextual bandits and episodic RL). However, it has long been known that TS is more effective empirically. For the cooperative RL setting, even though we do not know of any cooperative UCB results, we believe the same phenomenon will happen again. Note also that the O(\sqrt{nT}) scaling (as achieved by our cooperative TS) is optimal, so UCB cannot do any better theoretically. Additionally, TS has the advantage of being able to incorporate the prior much more effectively than UCB (which requires extensive tuning of the confidence width parameters), which is another practical advantage. As such, our paper's position is not to derive regret bounds so as to clarify which algorithm is better, but instead, take a practically well-performing algorithm that is widely known to have empirical efficacy (i.e. TS) and then provide theoretical guarantees for it.
>
> The above being said, during the rebuttal period, we have followed your advice on comparing our cooperative TS with cooperative UCB by performing more experiments, both for the finite-horizon setting and the infinite-horizon setting. We have added all the simulation results regarding comparing concurrent PSRL with concurrent UCB algorithm in appendix C (colored in blue) in the updated supplementary material (please refer there to see the new results). The simulation plots are Figure 3 and 4. From the simulations, we could see that the TS continues to outperform (widely) UCB in the cooperative setting. In our view, this outperformance comes not only from TS' single-agent effectiveness, but also from a compounding effect through diversity: concurrent UCB would have deterministic confidence bounds and hence assign the same policy to all agents (i.e. all agents would explore in exactly the same way), whereas our concurrent TS would drawn different policies for different agents based on the posterior. We believe the latter is an important merit of concurrent TS.

---

> > ### Comment · Reviewer_vsWP · 2022-08-05
> > **Response**
> >
> > Thanks for the feedbacks! I have read the revision and it is good to see that new simulation results validate the effectiveness of TS as compared to UCB. In particular, the experiments that explore the regret v.s. the number of agents partially verify the advantage of TS in terms of exploration when multiple data sources are available. While the theoretical analysis shows that TS does not enjoy statistical superiority as compared to UCB, I agree that providing theoretical guarantees for TS, which is known to be empirically more efficient, has its own values. Also, for the empirical evidence, you may include [1, 2] as the motivating examples, which also demonstrate the empirical superiority of TS.
> >
> > I still have a question on the advantage of TS about the parameter tuning. While I totally agree that tuning the confidence width of UCB is a   disaster, in the experiment, it seems that TS requires the knowledge of the types of underlying distribution for updating. This may not always hold. For another line of works of TS, focusing on the frequentist setting, the likelihood is simply an exponential weighted of the cumulative loss and we have no knowledge about the true prior and likelihood (e.g., [3]) and there is indeed some hyper-parameter to tune for the best performance. I am wondering in the bayesian setting, whether the knowledge of the prior type is critical. If we choose a wrong type that is different from the true one, will your algorithm still work?
> >
> > [1] Chapelle, Olivier, and Lihong Li. "An empirical evaluation of thompson sampling." Advances in neural information processing systems 24 (2011).
> >
> > [2] Osband, Ian, Benjamin Van Roy, and Zheng
> >
> > [3] Zhang, Tong. "Feel-good thompson sampling for contextual bandits and reinforcement learning." SIAM Journal on Mathematics of Data Science 4.2 (2022): 834-857.

---

> > > ### Author Response · Authors · 2022-08-06
> > > **Thank you very much for looking through the revised version!**
> > >
> > > We highly appreciate your continued efforts in reviewing our paper and the revised version, per your constructive comments. Also, thank you very much for bringing to our attention [1,2], which indeed provide good motivating examples for empirical evidence; and we will incorporate them in the introduction as further support of our approach. Thank you very much!
> > >
> > > Regarding the last point you brought up, we certainly agree that this is a general issue that every Bayesian algorithm faces, in that model mis-specification is bound to occur. In the literature, the typical flow is that regret guarantees on a promising algorithm is established in the no-model-mis-specification case before further investigations on model mis-specification is pursued. For instance, even in the simpler bandits setting, regret of TS is first proved on Bernoulli or Gaussian bandits (where the prior and likelihood are exactly as specified) before being further established on sub-Gaussian bandits (where the reward can be any sub-Gaussian distribution while the algorithm still assumes it is Gaussian).  As such, our view here is that 1) we acknowledge that this point you brought up is very valuable; and 2) even in the no model mis-specification setting, regret guarantee is not yet known for cooperative TS. In that sense, our results provide the first meaningful set of results that serve to catalyze further investigations on the much more difficult question of model mis-specification, which we leave for future work. We will mention this (including 3 you pointed out) in the conclusion.

---

> > > > ### Comment · Reviewer_vsWP · 2022-08-06
> > > > **Response**
> > > >
> > > > Thanks for clarification! Choosing an appropriate prior seems to be critical for TS as I also encounter this issue during my research. I believe that mentioning this point in the camera-ready version can make the status of the field clearer.
> > > >
> > > > I also suggest you to take a look at another line of work, focusing model-free approach and the frequentist (also referred to as the worst-case) regret bound: [3-6] where [4] considers the tabular MDP, [5] considers the linear MDP, [3] and [6] consider MDPs with general function approximation. From my side, model-based approach (like PSRL) may be limited to the problems where a realizable model class with moderate size is available and further study of model-free with frequentist regret bound is also promising. In particular, personally, for practical relevance, the idea of adding noise in the exploration [5] seems to be more promising as similar ideas have been employed in the empirical DRL study.
> > > >
> > > > Since my major concerns are addressed, I increase my score to 6 to support the acceptance of this work. Thanks again for all the responses.
> > > >
> > > > [4] Russo, Daniel. "Worst-case regret bounds for exploration via randomized value functions." Advances in Neural Information Processing Systems 32 (2019).
> > > >
> > > > [5] Zanette, Andrea, et al. "Frequentist regret bounds for randomized least-squares value iteration." International Conference on Artificial Intelligence and Statistics. PMLR, 2020.
> > > >
> > > > [6] Dann, Christoph, et al. "A provably efficient model-free posterior sampling method for episodic reinforcement learning." Advances in Neural Information Processing Systems 34 (2021): 12040-12051.

---

> > > > > ### Author Response · Authors · 2022-08-06
> > > > > **Thank you very much for the support and the valuable suggestions**
> > > > >
> > > > > We really appreciate your support and all the valuable suggestions. In the following we'll continue to explore the model-free based approach and also the frequentist regret bound results. Thank you also for all the valuable references you brought up on the frequentist regret side. We will incorporate all the references from 3 to 6 in our camera ready. Thank you again for bringing them to our attention, which certainly has helped improve the quality and comprehensiveness of the paper!

---

### Official Review · Reviewer_yBVn · 2022-07-10

**Rating:** 6
**Confidence:** 4
**Soundness:** 3 good
**Presentation:** 3 good
**Contribution:** 2 fair

**Summary:**

This work extends PSRL to the concurrent multi-agent setting. The authors discuss the finite horizon and infinite horizon settings, establishing a $1/\sqrt{n}$ per-agent regret, which is minimax optimal under a Dirichlet prior. For the infinite horizon case, the authors use a doubling schedule. In both settings the authors discuss results under general priors and under Dirichlet priors.

**Questions:**

Please see review above

**Ethics Review Area:**

["I don’t know"]

**Limitations:**

The authors didn't discuss negative societal impact (I'm unsure how relevant they are either).
Regarding limitations of their work, I didn't find a clear discussion.

**Strengths And Weaknesses:**

- Paper provides an extension of PSRL to the concurrent case, whereas previous work was based on UCRL-type algorithms.
- Proofs are provided for the theoretical results. The proofs seem correct, though clarity could be greatly improved.
- Please add the assumption that $r \in [0,1]$, even though it does not affect generality of the solution.
- I recommend stating all formal lemmas used from other papers in the appendix. This will greatly increase overall clarity of the proofs.
- It is hard to follow the proofs of the thoerems in the appendix. I recommend going over them too add some more explanations, and possibly divide them into a few more lemmas (if possible).
- Please provide an explicit algorithm in the appendix, where bonuses and posterior definitions are provided.
- Please find a way to put the simulations plots in the front matter - they should not be put in the appendix if they are referenced directly in the paper.
- How do the results in this paper generalize to the frequentist regret setting?
- The contribution of this paper is clear, but I am on the fence w.r.t. the overall strength of this paper. Particularly, theoretical aspects of the paper are standard and not surprising. The improvement the authors make over prior work is unclear. Overall, I worry that the extension the authors make is trivial w.r.t. current work.
- Simulations in this paper are very weak. There are many interesting aspects to test, especially in the frequentist setting, which is usually the relevant setting in real-world problems.

Comments regarding proof of Theorem 3.1 (Appendix):

- Line 519: where did this inequality come from?
- Lines 525-526: I understand what the authors wanted to say, but please rephrase this statement to make it more percise and clear.
- The use of lines 525-529, including Equations B.7 - B.10 is unclear in the final statement.
- It is unclear what statements the authors are using in the final equations of the proof.

---

> ### Author Response · Authors · 2022-08-02
> **Thank you for your positive review and constructive comments!**
>
> Thank you so much for your input. Please refer to the latest main pdf and supplementary materials (new results in blue and revised contents in red), where in particular we have added a new frequentist regret bound per your suggestion. Below are detailed responses (starting from your third point):
>
> 3. We added this reward assumption in section 2 —-  problem formulation under both finite-horizon and infinite-horizon settings, in subsection 2.1 and 2.2 respectively, colored in red.
>
> 4. We added useful lemmas from literature in appendix subsection D.1, G.1, colored in red. Things indeed now look more clear!
>
> 5. We reorganized the proofs and classified them into the corresponding groups according to which theorems/bounds they refer to: section D as the proof under finite-horizon case under general prior; section E as finite-horizon case under Dirichlet prior; section F as infinite-horizon case under general prior; section G as infinite-horizon case under Dirichlet prior. (colored in red).
>
> 6. Thank you for the suggestion! We restated algorithms in appendix section A and we also provided  the definitions of reward function posterior definitions. So on a high level, for the finite horizon case, we have n agents concurrently implement the PSRL algorithm, and they interact and gather information at the end of each episode, and then update their shared information set before the beginning of the new episode. For the infinite-horizon case, the n agents use doubling-epoch strategy to decide the break point for interaction, and then update their shared information set before continuing with the process. Appendix A also provides a detailed explanation of the reward functions and how we update the posterior distribution throughout the learning process (colored in blue).
>
> 7. We edited this part and moved the simulation plots of concurrent Thompson Sampling  to the main paper in section 5.
>
> 8. Thank you for bringing this up! We actually had a bound on this but decided to not include it at submission so as to have a more unifying theme of Bayesian regret. Per your suggestion, we have added it back (see the first 8 pages of the updated supplementary material), which now provides a worst-case (i.e. frequentist) bound for infinite-horizon case under Dirichlet Prior distribution. Due to page limitation and the purpose of keeping the consistency of the paper’s structure, we placed this part to appendix B for reference, colored in blue.
>
> 9. Our results provide the first regret bounds for concurrent TS. Additionally, in conjunction with the above mentioned changes, we hope you would favorably change your opinion on this.
>
> 10. During the rebuttal period, we have also strengthened the simulation results by comparing concurrent PSRL with concurrent UCB algorithm in appendix C (colored in blue). The simulation plots are Figure 3 and 4 in the updated supplementary materials. From the simulations, we could see that the TS continues to outperform (widely) UCB in the cooperative setting. In our view, this outperformance comes not only from TS' single-agent effectiveness, but also from a compounding effect through diversity: concurrent UCB would have deterministic confidence bounds and hence assign the same policy to all agents (i.e. all agents would explore in exactly the same way), whereas our concurrent TS would drawn different policies for different agents based on the posterior. We believe the latter is an important merit of concurrent TS.
>
> Minor comments:
> 1. This can be achieved by inserting delta=\frac{2}{Kn^2} into the formula of beta between line 622 and line 623 (before revision as line 517 and 518), and using the fact that K=\lfloor T/H\rfloor on line 623 (before revision as line 518)
>
> 2. Revised, line 632 to line 638, colored in red.
>
> 3. This is because in the proof we try to bound  (D.7) (previously B.6) on the event  of A_k(s,a) and the complement of event A_k(s,a), and then summing these two parts together. On event  A_k(s,a), (D.7) is bounded by the inequality between line 639 and line 640 (previously line 526 and line 527); and on the event of the complement of A_k(s,a), the key is to bound beta, which has an upper bound according to the inequality between line 624 and line 625 (previously line 519 and 520). This indicates that it’s sufficient to bound \sqrt{1/N_{t_k}(s_{kh}^p,\pi_{kp})}, summing across p, k and h. And in the end we sum these two parts together, to get the last part of the inequalities between line 644 and 645 (previously line 531 and 532).
>
> 4. This is because in order to bound the Bayesian regret, we need to bounded \delta_{kp} according to the regret decomposition derived in the earlier part of the proof of theorem 3.1. So according to the inequalities between 627 and 628, we need to bound (D.7). So that’s why providing a bound on (D.7) is sufficient to bound the Bayesian regret.
>
> Finally, we do not see any negative societal impact as the work is a theoretical study of the benefits of cooperation.

---

> > ### Comment · Reviewer_yBVn · 2022-08-08
> > **Response**
> >
> > Thank you, I am quite happy with the authors' improvements and have decided to increase my score to reflect this.

---

> > > ### Author Response · Authors · 2022-08-09
> > > **Thank you very much!**
> > >
> > > Thank you for your continued efforts in the review and for your support!

---

### Meta-Review · Area_Chair_Td4A · 2022-08-21

**Recommendation:** Accept
**Confidence:** Less certain

**Metareview:**

We thank the authors for their submission.

This well-written paper presents no-Bayesian-regret algorithms for multi-agent cooperative RL. This is the first work to study Thompson sampling algorithms in the context of concurrent RL - where agents interact with the MDP simultaneously. An experimental evaluation is also provided.

**Award:**

No

---

### Decision · Program_Chairs · 2022-09-14

Accept